# HOW DOES MIXUP HELP WITH ROBUSTNESS AND GENERALIZATION?

**Linjun Zhang**[*]
Rutgers University
linjun.zhang@rutgers.edu

**Zhun Deng**[*]
Harvard University
zhundeng@g.harvard.edu

**Kenji Kawaguchi**[*]
Harvard University
kkawaguchi@fas.harvard.edu

**Amirata Ghorbani**
Stanford University
amiratag@stanford.edu

**James Zou**
Stanford University
jamesz@stanford.edu

## ABSTRACT

Mixup is a popular data augmentation technique based on taking convex combinations of pairs of examples and their labels. This simple technique has been shown to substantially improve both the robustness and the generalization of the trained model. However, it is not well-understood why such improvement occurs. In this paper, we provide theoretical analysis to demonstrate how using Mixup in training helps model robustness and generalization. For robustness, we show that minimizing the Mixup loss corresponds to approximately minimizing an upper bound of the adversarial loss. This explains why models obtained by Mixup training exhibits robustness to several kinds of adversarial attacks such as Fast Gradient Sign Method (FGSM). For generalization, we prove that Mixup augmentation corresponds to a specific type of data-adaptive regularization which reduces overfitting. Our analysis provides new insights and a framework to understand Mixup.

## 1 INTRODUCTION

Mixup was introduced by Zhang et al. (2018) as a data augmentation technique. It has been empirically shown to substantially improve test performance and robustness to adversarial noise of state-of-the-art neural network architectures (Zhang et al., 2018; Lamb et al., 2019; Thulasidasan et al., 2019; Zhang et al., 2018; Arazo et al., 2019). Despite the impressive empirical performance, it is still not fully understood why Mixup leads to such improvement across the different aspects mentioned above. We first provide more background about robustness and generalization properties of deep networks and Mixup. Then we give an overview of our main contributions.

*Adversarial robustness.* Although neural networks have achieved remarkable success in many areas such as natural language processing (Devlin et al., 2018) and image recognition (He et al., 2016a), it has been observed that neural networks are very sensitive to adversarial examples — prediction can be easily flipped by human imperceptible perturbations (Goodfellow et al., 2014; Szegedy et al., 2013). Specifically, in Goodfellow et al. (2014), the authors use fast gradient sign method (FGSM) to generate adversarial examples, which makes an image of panda to be classified as gibbon with high confidence. Although various defense mechanisms have been proposed against adversarial attacks, those mechanisms typically sacrifice test accuracy in turn for robustness (Tsipras et al., 2018) and many of them require a significant amount of additional computation time. In contrast, Mixup training tends to improve test accuracy and at the same time also exhibits a certain degree of resistance to adversarial examples, such as those generated by FGSM (Lamb et al., 2019). Moreover, the corresponding training time is relatively modest. As an illustration, we compare the robust test

---

[*]Equal contribution.

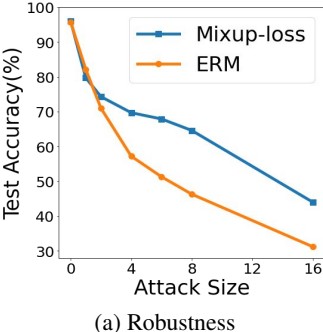 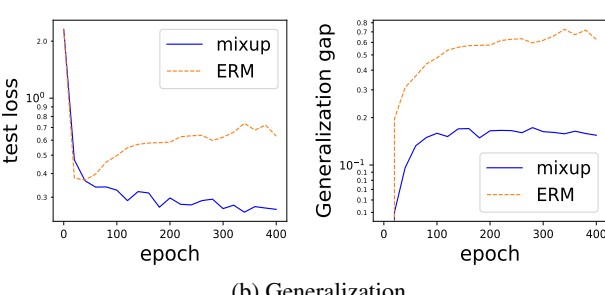

(a) Robustness         (b) Generalization

Figure 1: Illustrative examples of the impact of Mixup on robustness and generalization. (a) Adversarial robustness on the SVHN data under FGSM attacks. (b) Generalization gap between test and train loss. More details regarding the experimental setup are included in Appendix C.1, C.2.

accuracy between a model trained with Mixup and a model trained with standard empirical risk minimization (ERM) under adversarial attacks generated by FGSM (Fig. 1a). The model trained with Mixup loss has much better robust accuracy. Robustness of Mixup under other attacks have also been empirically studied in Lamb et al. (2019).

*Generalization.* Generalization theory has been a central focus of learning theory (Vapnik, 1979; 2013; Bartlett et al., 2002; Bartlett & Mendelson, 2002; Bousquet & Elisseeff, 2002; Xu & Mannor, 2012), but it still remains a mystery for many modern deep learning algorithms (Zhang et al., 2016; Kawaguchi et al., 2017). For Mixup, from Fig. (1b), we observe that Mixup training results in better test performance than the standard empirical risk minimization. That is mainly due to its good generalization property since the training errors are small for both Mixup training and empirical risk minimization (experiments with training error results are included in the appendix). While there have been many enlightening studies trying to establish generalization theory for modern machine learning algorithms (Sun et al., 2015; Neyshabur et al., 2015; Hardt et al., 2016; Bartlett et al., 2017; Kawaguchi et al., 2017; Arora et al., 2018; Neyshabur & Li, 2019), few existing studies have illustrated the generalization behavior of Mixup training in theory.

**Our contributions.** In this paper, we theoretically investigate how Mixup improves both adversarial robustness and generalization. We begin by relating the loss function induced by Mixup to the standard loss with additional adaptive regularization terms. Based on the derived regularization terms, we show that Mixup training minimizes an upper bound on the adversarial loss,which leads to the robustness against single-step adversarial attacks. For generalization, we show how the regularization terms can reduce over-fitting and lead to better generalization behaviors than those of standard training. Our analyses provides insights and framework to understand the impact of Mixup.

**Outline of the paper.** Section 2 introduces the notations and problem setup. In Section 3, we present our main theoretical results, including the regularization effect of Mixup and the subsequent analysis to show that such regularization improves adversarial robustness and generalization. Section 4 concludes with a discussion of future work. Proofs are deferred to the Appendix.

## 1.1 RELATED WORK

Since its advent, Mixup training (Zhang et al., 2018) has been shown to substantially improve generalization and single-step adversarial robustness among a wide rage of tasks, on both supervised (Lamb et al., 2019; Verma et al., 2019a; Guo et al., 2019), and semi-supervised settings (Berthelot et al., 2019; Verma et al., 2019b). This has motivated a recent line of work for developing a number of variants of Mixup, including Manifold Mixup (Verma et al., 2019a), Puzzle Mix (Kim et al., 2020), CutMix (Yun et al., 2019), Adversarial Mixup Resynthesis (Beckham et al., 2019), and PatchUp (Faramarzi et al., 2020). However, theoretical understanding of the underlying mechanism of why Mixup and its variants perform well on generalization and adversarial robustness is still limited.

Some of the theoretical tools we use in this paper are related to Wang & Manning (2013) and Wager et al. (2013), where the authors use second-order Taylor approximation to derive a regularized loss function for Dropout training. This technique is then extended to drive more properties of Dropout, including the inductive bias of Dropout (Helmbold & Long, 2015), the regularization effect in matrix factorization (Mianjy et al., 2018), and the implicit regularization in neural networks (Wei et al., 2020). This technique has been recently applied to Mixup in a parallel and independent work (Carratino et al., 2020) to derive regularization terms. Compared with the results in Carratino et al. (2020), our derived regularization enjoys a simpler form and therefore enables the subsequent analysis of adversarial robustness and generalization. We clarify the detailed differences in Section 3.

To the best of our knowledge, our paper is the first to provide a theoretical treatment to connect the regularization, adversarial robustness, and generalization for Mixup training.

## 2 PRELIMINARIES

In this section, we state our notations and briefly recap the definition of Mixup.

**Notations.** We denote the general parameterized loss as $l(\theta, z)$, where $\theta \in \Theta \subseteq \mathbb{R}^d$ and $z = (x, y)$ is the input and output pair. We consider a training dataset $S = \{(x_1, y_1), \cdots, (x_n, y_n)\}$, where $x_i \in \mathcal{X} \subseteq \mathbb{R}^p$ and $y_i \in \mathcal{Y} \subseteq \mathbb{R}^m$ are i.i.d. drawn from a joint distribution $\mathcal{P}_{x,y}$. We further denote $\tilde{x}_{i,j}(\lambda) = \lambda x_i + (1 - \lambda)x_j$, $\tilde{y}_{i,j}(\lambda) = \lambda y_i + (1 - \lambda)y_j$ for $\lambda \in [0, 1]$ and let $\tilde{z}_{i,j}(\lambda) = (\tilde{x}_{i,j}(\lambda), \tilde{y}_{i,j}(\lambda))$. Let $L(\theta) = \mathbb{E}_{z \sim \mathcal{P}_{x,y}} l(\theta, z)$ denote the standard population loss and $L_n^{std}(\theta, S) = \sum_{i=1}^n l(\theta, z_i)/n$ denote the standard empirical loss. For the two distributions $\mathcal{D}_1$ and $\mathcal{D}_2$, we use $p\mathcal{D}_1 + (1 - p)\mathcal{D}_2$ for $p \in (0, 1)$ to denote the mixture distribution such that a sample is drawn with probabilities $p$ and $(1 - p)$ from $\mathcal{D}_1$ and $\mathcal{D}_2$ respectively. For a parameterized function $f_\theta(x)$, we use $\nabla f_\theta(x)$ and $\nabla_\theta f_\theta(x)$ to respectively denote the gradient with respect to $x$ and $\theta$. For two vectors $a$ and $b$, we use $cos(x, y)$ to denote $\langle x, y \rangle / (\|x\| \cdot \|y\|)$.

**Mixup.** Generally, for classification cases, the output $y_i$ is the embedding of the class of $x_i$, *i.e.* the one-hot encoding by taking $m$ as the total number of classes and letting $y_i \in \{0, 1\}^m$ be the binary vector with all entries equal to zero except for the one corresponding to the class of $x_i$. In particular, if we take $m = 1$, it degenerates to the binary classification. For regression cases, $y_i$ can be any real number/vector. The Mixup loss is defined in the following form:

$$L_n^{\mathrm{mix}}(\theta, S) = \frac{1}{n^2} \sum_{i,j=1}^n \mathbb{E}_{\lambda \sim \mathcal{D}_\lambda} l(\theta, \tilde{z}_{ij}(\lambda)), \tag{1}$$

where $\mathcal{D}_\lambda$ is a distribution supported on $[0, 1]$. Throughout the paper, we consider the most commonly used $\mathcal{D}_\lambda$ – Beta distribution $Beta(\alpha, \beta)$ for $\alpha, \beta > 0$.

## 3 MAIN RESULTS

In this section, we first introduce a lemma that characterizes the regularization effect of Mixup. Based on this lemma, we then derive our main theoretical results on adversarial robustness and generalization error bound in Sections 3.2 and 3.3 respectively.

### 3.1 THE REGULARIZATION EFFECT OF MIXUP

As a starting point, we demonstrate how Mixup training is approximately equivalent to optimizing a regularized version of standard empirical loss $L_n^{std}(\theta, S)$. Throughout the paper, we consider the following class of loss functions for the prediction function $f_\theta(x)$ and target $y$:

$$\mathcal{L} = \{l(\theta, (x, y)) | l(\theta, (x, y)) = h(f_\theta(x)) - y f_\theta(x) \text{ for some function } h\}. \tag{2}$$

This function class $\mathcal{L}$ includes many commonly used losses, including the loss function induced by Generalized Linear Models (GLMs), such as linear regression and logistic regression, and also cross-entropy for neural networks. In the following, we introduce a lemma stating that the Mixup training with $\lambda \sim D_\lambda = Beta(\alpha, \beta)$ induces a regularized loss function with the weights of each regularization specified by a mixture of Beta distributions $\tilde{\mathcal{D}}_\lambda = \frac{\alpha}{\alpha+\beta} Beta(\alpha + 1, \beta) + \frac{\beta}{\alpha+\beta} Beta(\beta + 1, \alpha)$.

**Lemma 3.1.** *Consider the loss function $l(\theta, (x, y)) = h(f_\theta(x)) - y f_\theta(x)$, where $h(\cdot)$ and $f_\theta(\cdot)$ for all $\theta \in \Theta$ are twice differentiable. We further denote $\tilde{\mathcal{D}}_\lambda$ as a uniform mixture of two Beta distributions, i.e., $\frac{\alpha}{\alpha+\beta} Beta(\alpha+1, \beta) + \frac{\beta}{\alpha+\beta} Beta(\beta+1, \alpha)$, and $\mathcal{D}_X$ as the empirical distribution of the training dataset $S = (x_1, \cdots, x_n)$, the corresponding Mixup loss $L_n^{mix}(\theta, S)$, as defined in Eq. (1) with $\lambda \sim D_\lambda = Beta(\alpha, \beta)$, can be rewritten as*

$$L_n^{mix}(\theta, S) = L_n^{std}(\theta, S) + \sum_{i=1}^3 \mathcal{R}_i(\theta, S) + \mathbb{E}_{\lambda \sim \tilde{\mathcal{D}}_\lambda}[(1-\lambda)^2 \varphi(1-\lambda)],$$

*where $\lim_{a \to 0} \varphi(a) = 0$ and*

$$\mathcal{R}_1(\theta, S) = \frac{\mathbb{E}_{\lambda \sim \tilde{\mathcal{D}}_\lambda}[1-\lambda]}{n} \sum_{i=1}^n (h'(f_\theta(x_i)) - y_i) \nabla f_\theta(x_i)^\top \mathbb{E}_{r_x \sim \mathcal{D}_X}[r_x - x_i],$$

$$\mathcal{R}_2(\theta, S) = \frac{\mathbb{E}_{\lambda \sim \tilde{\mathcal{D}}_\lambda}[(1-\lambda)^2]}{2n} \sum_{i=1}^n h''(f_\theta(x_i)) \nabla f_\theta(x_i)^\top \mathbb{E}_{r_x \sim \mathcal{D}_X}[(r_x - x_i)(r_x - x_i)^\top] \nabla f_\theta(x_i),$$

$$\mathcal{R}_3(\theta, S) = \frac{\mathbb{E}_{\lambda \sim \tilde{\mathcal{D}}_\lambda}[(1-\lambda)^2]}{2n} \sum_{i=1}^n (h'(f_\theta(x_i)) - y_i) \mathbb{E}_{r_x \sim \mathcal{D}_X}[(r_x - x_i) \nabla^2 f_\theta(x_i)(r_x - x_i)^\top].$$

By putting the higher order terms of approximation in $\varphi(\cdot)$, this result shows that Mixup is related to regularizing $\nabla f_\theta(x_i)$ and $\nabla^2 f_\theta(x_i)$, which are the first and second directional derivatives with respect to $x_i$. Throughout the paper, our theory is mainly built upon analysis of the quadratic approximation of $L_n^{\text{mix}}(\theta, S)$, which we further denote as

$$\tilde{L}_n^{\text{mix}}(\theta, S) := L_n^{std}(\theta, S) + \sum_{i=1}^3 \mathcal{R}_i(\theta, S). \tag{3}$$

**Comparison with related work.** The result in Lemma 3.1 relies on the second-order Taylor expansion of the loss function Eq. (1). Similar approximations have been proposed before to study the regularization effect of Dropout training, see Wang & Manning (2013); Wager et al. (2013); Mianjy et al. (2018); Wei et al. (2020). Recently, Carratino et al. (2020) independently used similar approximation to study the regularization effect of Mixup. However, the regularization terms derived in Carratino et al. (2020) is much more complicated than those in Lemma 3.1. For example, in GLM, our technique yields the regularization term as shown in Lemma 3.3, which is much simpler than those in Corollaries 2 and 3 in Carratino et al. (2020). One technical step we use here to simplify the regularization expression is to equalize Mixup with input perturbation, see more details in the proof in the Appendix. This simpler expression enables us to study the robustness and generalization of Mixup in the subsequent sections.

**Validity of the approximation.** In the following, we present numerical experiments to support the approximation in Eq. (3). Following the setup of numerical validations in Wager et al. (2013); Carratino et al. (2020), we experimentally show that the quadratic approximation is generally very accurate. Specifically, we train a Logistic Regression model (as one example of a GLM model, which we study later) and a two layer neural network with ReLU activations. We use the two-moons dataset (Buitinck et al., 2013). Fig. 2 shows the training and test data's loss functions for training two models with different loss functions: the original Mixup loss and the approximate Mixup loss. Both models had the same random initialization scheme. Throughout training, we compute the test and training loss of each model using its own loss function. The empirical results shows the approximation of Mixup loss is quite close to the original Mixup loss.

## 3.2 MIXUP AND ADVERSARIAL ROBUSTNESS

Having introduced $\tilde{L}_n^{\text{mix}}(\theta, S)$ in Eq. (3), we are now ready to state our main theoretical results. In this subsection, we illustrate how Mixup helps adversarial robustness. We prove that minimizing $\tilde{L}_n^{\text{mix}}(\theta, S)$ is equivalent to minimizing an upper bound of the second order Taylor expansion of an adversarial loss.

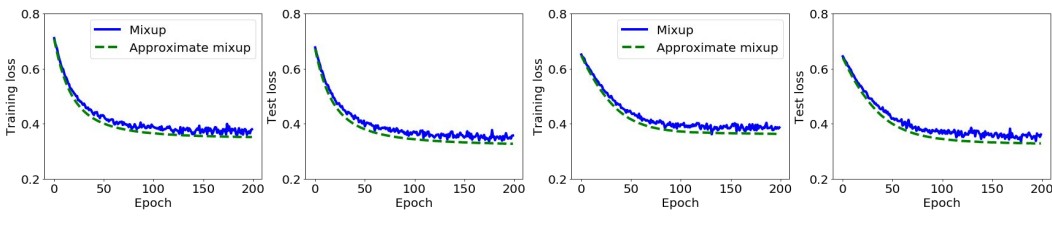

Figure 2: Comparison of the original Mixup loss with the approximate Mixup loss function.

Throughout this subsection, we study the logistic loss function

$$l(\theta, z) = \log(1 + \exp(f_\theta(x))) - y f_\theta(x),$$

where $y \in \mathcal{Y} = \{0, 1\}$. In addition, let $g$ be the logistic function such that $g(s) = e^s/(1 + e^s)$ and consider the case where $\theta$ is in the data-dependent space $\Theta$, defined as

$$\Theta = \{\theta \in \mathbb{R}^d : y_i f_\theta(x_i) + (y_i - 1) f_\theta(x_i) \geq 0 \text{ for all } i = 1, \ldots, n\}.$$

Notice that $\Theta$ contains the set of all $\theta$ with zero training errors:

$$\Theta \supseteq \{\theta \in \mathbb{R}^q : \text{the label prediction } \hat{y}_i = \mathbf{1}\{f_\theta(x_i) \geq 0\} \text{ is equal to } y_i \text{ for all } i = 1, \ldots, n\}. \quad (4)$$

In many practical cases, the training error (0-1 loss) becomes zero in finite time although the training loss does not. Equation (4) shows that the condition of $\theta \in \Theta$ is satisfied in finite time in such practical cases with zero training errors.

**Logistic regression.** As a starting point, we study the logistic regression with $f_\theta(x) = \theta^\top x$, in which case the number of parameters coincides with the data dimension, *i.e.* $p = d$. For a given $\varepsilon > 0$, we consider the adversarial loss with $\ell_2$-attack of size $\varepsilon\sqrt{d}$, that is, $L_n^{adv}(\theta, S) = 1/n \sum_{i=1}^n \max_{\|\delta_i\|_2 \leq \varepsilon\sqrt{d}} l(\theta, (x_i + \delta_i, y_i))$. We first present the following second order Taylor approximation of $L_n^{adv}(\theta, S)$.

**Lemma 3.2.** *The second order Taylor approximation of $L_n^{adv}(\theta, S)$ is $\sum_{i=1}^n \tilde{l}_{adv}(\varepsilon\sqrt{d}, (x_i, y_i))/n$, where for any $\eta > 0$, $x \in \mathbb{R}^p$ and $y \in \{0, 1\}$,*

$$\tilde{l}_{adv}(\eta, (x, y)) = l(\theta, (x, y)) + \eta |g(x^\top \theta) - y| \cdot \|\theta\|_2 + \frac{\eta^2}{2} \cdot g(x^\top \theta)(1 - g(x^\top \theta)) \cdot \|\theta\|_2^2. \quad (5)$$

By comparing $\tilde{l}_{adv}(\delta, (x, y))$ and $\tilde{L}_n^{\text{mix}}(\theta, S)$ applied to logistic regression, we prove the following.

**Theorem 3.1.** *Suppose that $f_\theta(x) = x^\top \theta$ and there exists a constant $c_x > 0$ such that $\|x_i\|_2 \geq c_x\sqrt{d}$ for all $i \in \{1, \ldots, n\}$. Then, for any $\theta \in \Theta$, we have*

$$\tilde{L}_n^{mix}(\theta, S) \geq \frac{1}{n} \sum_{i=1}^n \tilde{l}_{adv}(\varepsilon_i \sqrt{d}, (x_i, y_i)) \geq \frac{1}{n} \sum_{i=1}^n \tilde{l}_{adv}(\varepsilon_{\text{mix}} \sqrt{d}, (x_i, y_i))$$

*where $\varepsilon_i = R_i c_x \mathbb{E}_{\lambda \sim \tilde{\mathcal{D}}_\lambda}[1 - \lambda]$ with $R_i = |\cos(\theta, x_i)|$, and $\varepsilon_{\text{mix}} = R \cdot c_x \mathbb{E}_{\lambda \sim \tilde{\mathcal{D}}_\lambda}[1 - \lambda]$ with $R = \min_{i \in \{1, \ldots, n\}} |\cos(\theta, x_i)|$.*

Theorem 3.1 suggests that $\tilde{L}_n^{\text{mix}}(\theta, S)$ is an upper bound of the second order Taylor expansion of the adversarial loss with $\ell_2$-attack of size $\varepsilon_{\text{mix}}\sqrt{d}$. Note that $\varepsilon_{\text{mix}}$ depends on $\theta$; one can think the final radius is taken at the minimizer of $\tilde{L}_n^{\text{mix}}(\theta, S)$. Therefore, minimizing the Mixup loss would result in a small adversarial loss. Our analysis suggests that Mixup by itself can improve robustness against small attacks, which tend to be single-step attacks (Lamb et al., 2019). An interesting direction for future work is to explore whether combining Mixup with adversarial training is able to provide robustness against larger and more sophisticated attacks such as iterative projected gradient descent and other multiple-step attacks.

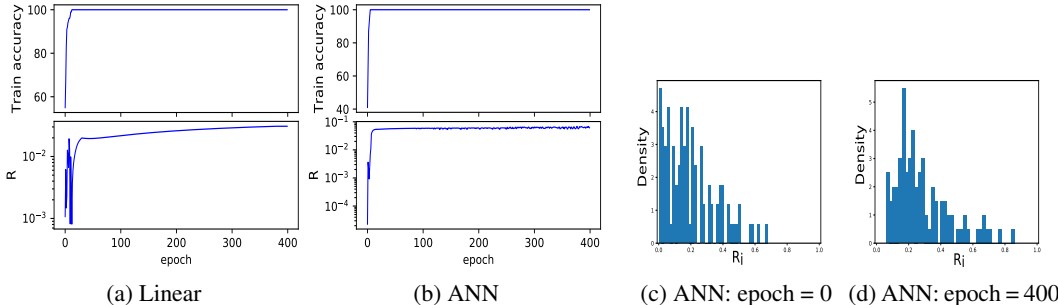

Figure 3: The behaviors of the values of $R$ and $R_i$ during training for linear models and artificial neural network with ReLU (ANN). The subplots (c) and (d) show the histogram of $(R_1, R_2, \ldots, R_n)$ for ANN before and after training. $R$ and $R_i$ control the radii of adversarial attacks that Mixup training protects for.

**Remark 3.1.** *Note that Theorem 3.1 also implies adversarial robustness against $\ell_\infty$ attacks with size $\varepsilon$ since for any attack $\delta$, $\|\delta\|_\infty \leq \varepsilon$ implies $\|\delta\|_2 \leq \varepsilon\sqrt{d}$, and therefore $\max_{\|\delta\|_\infty \leq \epsilon} l(\theta, (x+\delta, y)) \leq \max_{\|\delta\|_2 \leq \sqrt{d}\cdot\epsilon} l(\theta, (x+\delta, y))$.*

In the following we provide more discussion about the range of $R = \min_{i \in \{1,\ldots,n\}} |\cos(\theta, x_i)|$. We first show that under additional regularity conditions, we can obtain a high probability lower bound that does not depend on sample size. We then numerically demonstrate that $R$ tends to increase during training for both cases of linear models and neural networks at the end of this subsection.

*A constant lower bound for logistic regression.* Now, we show how to obtain a constant lower bound by adding some additional conditions.

**Assumption 3.1.** *Let us denote $\hat{\Theta}_n \subseteq \Theta$ as the set of minimizers of $\tilde{L}_n^{mix}(\theta, S)$. We assume there exists a set $\Theta^*$ [1], such that for all $n \geq N$, where $N$ is a positive integer, $\hat{\Theta}_n \subseteq \Theta^*$ with probability at least $1 - \delta_n$ where $\delta_n \to 0$ as $n \to 0$. Moreover, there exists a $\tau \in (0, 1)$ such that*

$$p_\tau = \mathbb{P}\left(\{x \in \mathcal{X} : |\cos(x, \theta)| \geq \tau \text{ for all } \theta \in \Theta^*\}\right) \in (0, 1].$$

Such condition generally holds for regular optimization problems, where the minimizers are not located too dispersedly in the sense of solid angle (instead of Euclidean distance). More specifically, if we normalize all the minimizers' $\ell_2$ norm to 1, this assumption requires that the set of minimizers should not be located all over the sphere. In addition, Assumption 3.1 only requires that the probability $p_\tau$ and the threshold $\tau$ to be non-zero. In particular, if the distribution of $x$ has positive mass in all solid angles, then when the set of minimizers is discrete, this assumption holds. For more complicated cases in which the set of minimizers consists of sub-manifolds, as long as there exists a solid angle in $\mathcal{X}$ that is disjoint with the set of minimizers, the assumption still holds.

**Theorem 3.2.** *Under Assumption 3.1, for $f_\theta(x) = x^\top \theta$, if there exists constants $b_x, c_x > 0$ such that $c_x\sqrt{d} \leq \|x_i\|_2 \leq b_x\sqrt{d}$ for all $i \in \{1, \ldots, n\}$. Then, with probability at least $1 - \delta_n - 2\exp(-np_\tau^2/2)$, there exists constants $\kappa > 0, \kappa_2 > \kappa_1 > 0$, such that for any $\theta \in \hat{\Theta}_n$, we have*

$$\tilde{L}_n^{mix}(\theta, S) \geq \frac{1}{n}\sum_{i=1}^n \tilde{l}_{adv}(\tilde{\varepsilon}_{\text{mix}}\sqrt{d}, (x_i, y_i))$$

*where $\tilde{\varepsilon}_{\text{mix}} = \tilde{R}c_x\mathbb{E}_{\lambda\sim\tilde{\mathcal{D}}_\lambda}[1-\lambda]$ and $\tilde{R} = \min\left\{\frac{p_\tau\kappa_1}{2\kappa_2 - p_\tau(\kappa_2-\kappa_1)}, \sqrt{\frac{4\kappa p_\tau}{2-p_\tau+4\kappa p_\tau}}\right\}\tau$.*

**Neural networks with ReLU / Max-pooling.** The results in the above subsection can be extended to the case of neural networks with ReLU activation functions and max-pooling. Specifically, we

---

[1]Under some well-separation and smoothness conditions, we would expect all elements in $\hat{\Theta}_n$ will fall into a neighborhood $\mathcal{N}_n$ of minimizers of $\mathbb{E}_S \tilde{L}_n^{\text{mix}}(\theta, S)$, and $\mathcal{N}_n$ will shrink as $n$ increases, i.e. $\mathcal{N}_{n+1} \subset \mathcal{N}_n$. One can think $\Theta^*$ is a set containing all $\mathcal{N}_n$ for $n \geq N$.

consider the logistic loss, $l(\theta, z) = \log(1 + \exp(f_\theta(x))) - y f_\theta(x)$ with $y \in \{0, 1\}$, where $f_\theta(x)$ represents a fully connected neural network with ReLU activation function or max-pooling:

$$f_\theta(x) = \beta^\top \sigma(W_{N-1} \cdots (W_2 \sigma(W_1 x))).$$

Here, $\sigma$ represents nonlinearity via ReLU and max pooling, each $W_i$ is a matrix, and $\beta$ is a column vector: i.e., $\theta$ consists of $\{W_i\}_{i=1}^{N-1}$ and $\beta$. With the nonlinearity $\sigma$ for ReLU and max-pooling, the function $f_\theta$ satisfies that $f_\theta(x) = \nabla f_\theta(x)^\top x$ and $\nabla^2 f_\theta(x) = 0$ almost everywhere, where the gradient is taken with respect to input $x$. Under such conditions, similar to Lemma 3.2, the adversarial loss function $\sum_{i=1}^n \max_{\|\delta_i\|_2 \leq \varepsilon \sqrt{d}} l(\theta, (x_i + \delta_i, y_i))/n$ can be written as

$$L_n^{std}(\theta, S) + \varepsilon_{\text{mix}} \sqrt{d} \left( \frac{1}{n} \sum_{i=1}^n |g(f_\theta(x_i)) - y_i| \|\nabla f_\theta(x_i)\|_2 \right) + \frac{\varepsilon_{\text{mix}}^2 d}{2} \left( \frac{1}{n} \sum_{i=1}^n |h''(f_\theta(x_i))| \|\nabla f_\theta(x_i)\|_2^2 \right).$$

(6)

With a little abuse of notations, we also denote

$$\tilde{l}_{adv}(\delta, (x, y)) = l(\theta, (x, y)) + \delta |g(f_\theta(x)) - y| \|\nabla f_\theta(x)\|_2 + \frac{\delta^2 d}{2} |h''(f_\theta(x))| \|\nabla f_\theta(x)\|_2^2.$$

The following theorem suggests that minimizing the Mixup loss in neural nets also lead to a small adversarial loss.

**Theorem 3.3.** *Assume that $f_\theta(x_i) = \nabla f_\theta(x_i)^\top x_i$, $\nabla^2 f_\theta(x_i) = 0$ (which are satisfied by the ReLU and max-pooling activation functions) and there exists a constant $c_x > 0$ such that $\|x_i\|_2 \geq c_x \sqrt{d}$ for all $i \in \{1, \ldots, n\}$. Then, for any $\theta \in \Theta$, we have*

$$\tilde{L}_n^{mix}(\theta, S) \geq \frac{1}{n} \sum_{i=1}^n \tilde{l}_{adv}(\varepsilon_i \sqrt{d}, (x_i, y_i)) \geq \frac{1}{n} \sum_{i=1}^n \tilde{l}_{adv}(\varepsilon_{\text{mix}} \sqrt{d}, (x_i, y_i))$$

*where $\varepsilon_i = R_i c_x \mathbb{E}_{\lambda \sim \tilde{\mathcal{D}}_\lambda}[1 - \lambda]$, $\varepsilon_{\text{mix}} = R \cdot c_x \mathbb{E}_{\lambda \sim \tilde{\mathcal{D}}_\lambda}[1 - \lambda]$ and $R_i = |\cos(\nabla f_\theta(x_i), x_i)|$, $R = \min_{i \in \{1, \ldots, n\}} |\cos(\nabla f_\theta(x_i), x_i)|$.*

Similar constant lower bound can be derived to the setting of neural networsk. Due to limited space, please see the detailed discussion in the appendix.

*On the value of $R = \min_i R_i$ via experiments.* For both linear models and neural networks, after training accuracy reaches 100%, the logistic loss is further minimized when $\|f_\theta(x_i)\|_2$ increases. Since $\|f_\theta(x_i)\|_2 = \|\nabla f_\theta(x_i)^\top x_i\|_2 = \|\nabla f_\theta(x_i)\|_2 \|x_i\|_2 R_i$, this suggests that $R_i$ and $R$ tend to increase after training accuracy reaches 100% (e.g., $\nabla f_\theta(x_i) = \theta$ in the case of linear models). We confirm this phenomenon in Fig. 3. In the figure, $R$ is initially small but tends to increase after training accuracy reaches 100%, as expected. For example, for ANN, the value of $R$ was initially $2.27 \times 10^{-5}$ but increased to $6.11 \times 10^{-2}$ after training. Fig. 3 (c) and (d) also show that $R_i$ for each $i$-th data point tends to increase during training and that the values of $R_i$ for many points are much larger than the pessimistic lower bound $R$: e.g., whereas $R = 6.11 \times 10^{-2}$, we have $R_i > 0.8$ for several data points in Fig. 3 (d). For this experiment, we generated 100 data points as $x_i \sim \mathcal{N}(0, I)$ and $y_i = \mathbb{1}\{x_i^\top \theta^* > 0\}$ where $x_i \in \mathbb{R}^{10}$ and $\theta^* \sim \mathcal{N}(0, I)$. We used SGD to train linear models and ANNs with ReLU activations and 50 neurons per each of two hidden layers. We set the learning rate to be 0.1 and the momentum coefficient to be 0.9. We turned off weight decay so that $R$ is not maximized as a result of bounding $\|\nabla f_\theta(x_i)\|$, which is a trivial case from the discussion above.

### 3.3 MIXUP AND GENERALIZATION

In this section, we show that the data-dependent regularization induced by Mixup directly controls the Rademacher complexity of the underlying function classes, and therefore yields concrete generalization error bounds. We study two models – the Generalized Linear Model (GLM) and two-layer ReLU nets with squared loss.

**Generalized linear model.** A Generalized Linear Model is a flexible generalization of ordinary linear regression, where the corresponding loss takes the following form:

$$l(\theta, (x, y)) = A(\theta^\top x) - y \theta^\top x,$$

where $A(\cdot)$ is the log-partition function, $x \in \mathbb{R}^p$ and $y \in \mathbb{R}$. For instance, if we take $A(\theta^\top x) = \log(1 + e^{\theta^\top x})$ and $y \in \{0, 1\}$, then the model corresponds to the logistic regression. In this paragraph, we consider the case where $\Theta$, $\mathcal{X}$ and $\mathcal{Y}$ are all bounded.

By further taking advantage of the property of shift and scaling invariance of GLM, we can further simplify the regularization terms in Lemma 3.1 and obtain the following results.

**Lemma 3.3.** *Consider the centralized dataset $S$, that is, $1/n \sum_{i=1}^n x_i = 0$. and denote $\hat{\Sigma}_X = \frac{1}{n} x_i x_i^\top$. For a GLM, if $A(\cdot)$ is twice differentiable, then the regularization term obtained by the second-order approximation of $\tilde{L}_n^{mix}(\theta, S)$ is given by*

$$\frac{1}{2n} [\sum_{i=1}^n A''(\theta^\top x_i)] \cdot \mathbb{E}_{\lambda \sim \tilde{\mathcal{D}}_\lambda} [\frac{(1-\lambda)^2}{\lambda^2}] \theta^\top \hat{\Sigma}_X \theta, \tag{7}$$

*where $\tilde{\mathcal{D}}_\lambda = \frac{\alpha}{\alpha+\beta} Beta(\alpha + 1, \beta) + \frac{\alpha}{\alpha+\beta} Beta(\beta + 1, \alpha)$.*

Given the above regularization term, we are ready to investigate the corresponding generalization gap. Following similar approaches in Arora et al. (2020), we shed light upon the generalization problem by investigating the following function class that is closely related to the dual problem of Eq. (7):

$$\mathcal{W}_\gamma := \{x \to \theta^\top x, \text{ such that } \theta \text{ satisfying } \mathbb{E}_x A''(\theta^\top x) \cdot \theta^\top \Sigma_X \theta \leq \gamma\},$$

where $\alpha > 0$ and $\Sigma_X = \mathbb{E}[x x^\top]$. Further, we assume that the distribution of $x$ is $\rho$-*retentive* for some $\rho \in (0, 1/2]$, that is, if for any non-zero vector $v \in \mathbb{R}^d$, $[\mathbb{E}_x[A''(x^\top v)]]^2 \geq \rho \cdot \min\{1, \mathbb{E}_x(v^\top x)^2\}$. Such an assumption has been similarly assumed in Arora et al. (2020) and is satisfied by general GLMs when $\theta$ has bounded $\ell_2$ norm. We then have the following theorem.

**Theorem 3.4.** *Assume that the distribution of $x_i$ is $\rho$-retentive, and let $\Sigma_X = \mathbb{E}[x x^\top]$. Then the empirical Rademacher complexity of $\mathcal{W}_\gamma$ satisfies*

$$Rad(\mathcal{W}_\gamma, S) \leq \max\{(\frac{\gamma}{\rho})^{1/4}, (\frac{\gamma}{\rho})^{1/2}\} \cdot \sqrt{\frac{rank(\Sigma_X)}{n}}.$$

The above bound on Rademacher complexity directly implies the following generalization gap of Mixup training.

**Corollary 3.1.** *Suppose $A(\cdot)$ is $L_A$-Lipchitz continuous, $\mathcal{X}$, $\mathcal{Y}$ and $\Theta$ are all bounded, then there exists constants $L, B > 0$, such that for all $\theta$ satisfying $\mathbb{E}_x A''(\theta^\top x) \cdot \theta^\top \Sigma_X \theta \leq \gamma$ (the regularization induced by Mixup), we have*

$$L(\theta) \leq L_n^{std}(\theta, S) + 2L \cdot L_A \cdot \left( \max\{(\frac{\gamma}{\rho})^{1/4}, (\frac{\gamma}{\rho})^{1/2}\} \cdot \sqrt{\frac{rank(\Sigma_X)}{n}} \right) + B\sqrt{\frac{\log(1/\delta)}{2n}},$$

*with probability at least $1 - \delta$.*

**Remark 3.2.** *This result shows that the Mixup training would adapt to the intrinsic dimension of $x$ and therefore has a smaller generalization error. Specifically, if we consider the general ridge penalty and consider the function class $\mathcal{W}_\gamma^{ridge} := \{x \to \theta^\top x, \|\theta\|^2 \leq \gamma\}$, then the similar technique would yield a Rademacher complexity bound $Rad(\mathcal{W}_\gamma, S) \leq \max\{(\gamma/\rho)^{1/4}, (\gamma/\rho)^{1/2}\} \cdot \sqrt{p/n}$, where $p$ is the dimension of $x$. This bound is much larger than the result in Theorem 3.4 when the intrinsic dimension $rank(\Sigma_X)$ is small.*

**Non-linear cases.** The above results on GLM can be extended to the non-linear neural network case with Manifold Mixup (Verma et al., 2019a). In this section, we consider the two-layer ReLU neural networks with the squared loss $L(\theta, S) = \frac{1}{n} \sum_{i=1}^n (y_i - f_\theta(x_i))^2$, where $y \in \mathbb{R}$ and $f_\theta(x)$ is a two-layer ReLU neural network, with the form of

$$f_\theta(x) = \theta_1^\top \sigma(W x) + \theta_0.$$

where $W \in \mathbb{R}^{p \times d}$, $\theta_1 \in \mathbb{R}^d$, and $\theta_0$ denotes the bias term. Here, $\theta$ consists of $W$, $\theta_0$ and $\theta_1$.

If we perform Mixup on the second layer (i.e., mix neurons on the hidden layer as proposed by Verma et al. (2019a)), we then have the following result on the induced regularization.

**Lemma 3.4.** *Denote $\hat{\Sigma}_X^\sigma$ as the sample covariance matrix of $\{\sigma(Wx_i)\}_{i=1}^n$, then the regularization term obtained by the second-order approximation of $\tilde{L}_n^{mix}(\theta, S)$ is given by*

$$\mathbb{E}_{\lambda \sim \tilde{\mathcal{D}}_\lambda}[\frac{(1-\lambda)^2}{\lambda^2}]\theta_1^\top \hat{\Sigma}_X^\sigma \theta_1, \tag{8}$$

*where $\tilde{\mathcal{D}}_\lambda \sim \frac{\alpha}{\alpha+\beta}Beta(\alpha+1, \beta) + \frac{\beta}{\alpha+\beta}Beta(\beta+1, \alpha)$.*

To show the generalization property of this regularizer, similar to the last section, we consider the following distribution-dependent class of functions indexed by $\theta$:

$$\mathcal{W}_\gamma^{NN} := \{x \to f_\theta(x), \text{ such that } \theta \text{ satisfying } \theta_1^\top \Sigma_X^\sigma \theta_1 \leqslant \gamma\},$$

where $\Sigma_X^\sigma = \mathbb{E}[\hat{\Sigma}_X^\sigma]$ and $\alpha > 0$. We then have the following result.

**Theorem 3.5.** *Let $\mu_\sigma = \mathbb{E}[\sigma(Wx)]$ and denote the generalized inverse of $\Sigma_X^\sigma$ by $\Sigma_X^{\sigma\dagger}$. Suppose $\mathcal{X}$, $\mathcal{Y}$ and $\Theta$ are all bounded, then there exists constants $L, B > 0$, such that for all $f_\theta$ in $\mathcal{W}_\gamma^{NN}$ (the regularization induced by Manifold Mixup), we have, with probability at least $1 - \delta$,*

$$L(\theta) \leqslant L_n^{std}(\theta, S) + 4L \cdot \sqrt{\frac{\gamma \cdot (rank(\Sigma_X^\sigma) + \|\Sigma_X^{\sigma\dagger/2}\mu_\sigma\|^2)}{n}} + B\sqrt{\frac{\log(1/\delta)}{2n}}.$$

## 4 CONCLUSION AND FUTURE WORK

Mixup is a data augmentation technique that generates new samples by linear interpolation of multiple samples and their labels. The Mixup training method has been empirically shown to have better generalization and robustness against attacks with adversarial examples than the traditional training method, but there is a lack of rigorous theoretical understanding. In this paper, we prove that the Mixup training is approximately a regularized loss minimization. The derived regularization terms are then used to demonstrate why Mixup has improved generalization and robustness against one-step adversarial examples. One interesting future direction is to extend our analysis to other Mixup variants, for example, Puzzle Mix (Kim et al., 2020) and Adversarial Mixup Resynthesis (Beckham et al., 2019), and investigate if the generalization performance and adversarial robustness can be further improved by these newly developed Mixup methods.

### ACKNOWLEDGMENTS

The research of Linjun Zhang is supported by NSF DMS-2015378. The research of James Zou is supported by NSF CCF 1763191, NSF CAREER 1942926 and grants from the Silicon Valley Foundation and the Chan-Zuckerberg Initiative. The research of Kenji Kawaguchi is partially supported by the Center of Mathematical Sciences and Applications at Harvard University. This work is also in part supported by NSF award 1763665.

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

# Appendix

In this appendix, we provide proofs of the main theorems and the corresponding technical lemmas. Additional discussion on the range of $R$ in the case of neural nets, and some further numerical experiments are also provided.

## A  TECHNIQUE PROOFS

### A.1  PROOF OF LEMMA 3.1

Consider the following problem with loss function $l_{x,y}(\theta) := l(\theta, (x, y)) = h(f_\theta(x)) - y f_\theta(x)$, that is

$$L_n^{std}(\theta, S) = \frac{1}{n} \sum_{i=1}^{n} [h(f_\theta(x_i)) - y_i f_\theta(x_i)].$$

The corresponding Mixup version, as defined in Eq.(1), is

$$L_n^{mix}(\theta, S) = \frac{1}{n^2} \mathbb{E}_{\lambda \sim Beta(\alpha, \beta)} \sum_{i,j=1}^{n} [h(f_\theta(\tilde{x}_{i,j}(\lambda))) - (\lambda y_i + (1-\lambda) y_j) f_\theta(\tilde{x}_{i,j}(\lambda))],$$

where $\tilde{x}_{i,j}(\lambda) = \lambda x_i + (1-\lambda) x_j$. Further transformation leads to

$$
\begin{aligned}
L_n^{mix}(\theta, S) &= \frac{1}{n^2} \mathbb{E}_{\lambda \sim Beta(\alpha, \beta)} \sum_{i,j=1}^{n} \Big\{ \lambda h(f_\theta(\tilde{x}_{i,j}(\lambda)))) - \lambda y_i f_\theta(\tilde{x}_{i,j}(\lambda)) \\
&\quad + (1-\lambda) h(f_\theta(\tilde{x}_{i,j}(\lambda))) - (1-\lambda) y_j f_\theta(\tilde{x}_{i,j}(\lambda)) \Big\} \\
&= \frac{1}{n^2} \mathbb{E}_{\lambda \sim Beta(\alpha, \beta)} \mathbb{E}_{B \sim Bern(\lambda)} \sum_{i,j=1}^{n} \Big\{ B[h(f_\theta(\tilde{x}_{i,j}(\lambda))) - y_i f_\theta(\tilde{x}_{i,j}(\lambda))] \\
&\quad + (1-B)[h(f_\theta(\tilde{x}_{i,j}(\lambda))) - y_j f_\theta(\tilde{x}_{i,j}(\lambda))] \Big\}
\end{aligned}
$$

Note that $\lambda \sim Beta(\alpha, \beta), B|\lambda \sim Bern(\lambda)$, by conjugacy, we can exchange them in order and have

$$B \sim Bern(\frac{\alpha}{\alpha + \beta}), \lambda \mid B \sim Beta(\alpha + B, \beta + 1 - B).$$

As a result,

$$
\begin{aligned}
L_n^{mix}(\theta, S) &= \frac{1}{n^2} \sum_{i,j=1}^{n} \Big\{ \frac{\alpha}{\alpha + \beta} \mathbb{E}_{\lambda \sim Beta(\alpha+1, \beta)} [h(f_\theta(\tilde{x}_{i,j}(\lambda))) - y_i f_\theta(\tilde{x}_{i,j}(\lambda))] \\
&\quad + \frac{\beta}{\alpha + \beta} \mathbb{E}_{\lambda \sim Beta(\alpha, \beta+1)} [h(f_\theta(\tilde{x}_{i,j}(\lambda))) - y_j f_\theta(\tilde{x}_{i,j}(\lambda))] \Big\}.
\end{aligned}
$$

Using the fact $1 - Beta(\alpha, \beta+1)$ and $Beta(\beta+1, \alpha)$ are of the same distribution and $\tilde{x}_{ij}(1-\lambda) = \tilde{x}_{ji}(\lambda)$, we have

$$
\begin{aligned}
&\sum_{i,j} \mathbb{E}_{\lambda \sim Beta(\alpha, \beta+1)} [h(f_\theta(\tilde{x}_{i,j}(\lambda))) - y_j f_\theta(\tilde{x}_{i,j}(\lambda))] \\
&= \sum_{i,j} \mathbb{E}_{\lambda \sim Beta(\beta+1, \alpha)} [h(f_\theta(\tilde{x}_{i,j}(\lambda))) - y_i f_\theta(\tilde{x}_{i,j}(\lambda))].
\end{aligned}
$$

Thus, let $\tilde{\mathcal{D}}_\lambda = \frac{\alpha}{\alpha+\beta} Beta(\alpha+1, \beta) + \frac{\beta}{\alpha+\beta} Beta(\beta+1, \alpha)$

$$
\begin{aligned}
L_n^{mix}(\theta, S) &= \frac{1}{n} \sum_{i=1}^{n} \mathbb{E}_{\lambda \sim \tilde{\mathcal{D}}_\lambda} \mathbb{E}_{r_x \sim D_x} h(f(\theta, \lambda x_i + (1-\lambda) r_x))) - y_i f(\theta, \lambda x_i + (1-\lambda) r_x) \\
&= \frac{1}{n} \sum_{i=1}^{n} \mathbb{E}_{\lambda \sim \tilde{\mathcal{D}}_\lambda} \mathbb{E}_{r_x \sim D_x} l_{\tilde{x}_i, y_i}(\theta) \quad\quad (9)
\end{aligned}
$$

where $D_x$ is the empirical distribution induced by training samples, and $\check{x}_i = \lambda x_i + (1 - \lambda)r_x$.

In the following, denote $\check{S} = \{(\check{x}_i, y_i)\}_{i=1}^n$, and let us analyze $L_n^{std}(\theta, \check{S}) = \frac{1}{n}\sum_{i=1}^n l_{\check{x}_i, y_i}(\theta)$, and compare it with $L_n^{std}(\theta, S)$. Let $\alpha = 1 - \lambda$ and $\psi_i(\alpha) = l_{\check{x}_i, y_i}(\theta)$. Then, using the definition of the twice-differentiability of function $\psi_i$,

$$l_{\check{x}_i, y_i}(\theta) = \psi_i(\alpha) = \psi_i(0) + \psi_i'(0)\alpha + \frac{1}{2}\psi_i''(0)\alpha^2 + \alpha^2\varphi_i(\alpha), \tag{10}$$

where $\lim_{z \to 0} \varphi_i(z) = 0$. By linearity and chain rule,

$$\psi_i'(\alpha) = h'(f_\theta(\check{x}_i))\frac{\partial f_\theta(\check{x}_i)}{\partial \check{x}_i}\frac{\partial \check{x}_i}{\partial \alpha} - y_i\frac{\partial f_\theta(\check{x}_i)}{\partial \check{x}_i}\frac{\partial \check{x}_i}{\partial \alpha}$$

$$= h'(f_\theta(\check{x}_i))\frac{\partial f_\theta(\check{x}_i)}{\partial \check{x}_i}(r_x - x_i) - y_i\frac{\partial f_\theta(\check{x}_i)}{\partial \check{x}_i}(r_x - x_i)$$

where we used $\frac{\partial \check{x}_i}{\partial \alpha} = (r_x - x_i)$. Since

$$\frac{\partial}{\partial \alpha}\frac{\partial f_\theta(\check{x}_i)}{\partial \check{x}_i}(r_x - x_i) = \frac{\partial}{\partial \alpha}(r_x - x_i)^\top[\frac{\partial f_\theta(\check{x}_i)}{\partial \check{x}_i}]^\top = (r_x - x_i)^\top\nabla^2 f_\theta(\check{x}_i)\frac{\partial \check{x}_i}{\partial \alpha} = (r_x - x_i)^\top\nabla^2 f_\theta(\check{x}_i)(r_x - x_i),$$

we have

$$\psi_i''(\alpha) = h'(f_\theta(\check{x}_i))(r_x - x_i)^\top\nabla^2 f_\theta(\check{x}_i)(r_x - x_i)$$
$$+ h''(f_\theta(\check{x}_i))[\frac{\partial f_\theta(\check{x}_i)}{\partial \check{x}_i}(r_x - x_i)]^2 - y_i(r_x - x_i)^\top\nabla^2 f_\theta(\check{x}_i)(r_x - x_i).$$

Thus,

$$\psi_i'(0) = h'(f_\theta(x_i))\nabla f_\theta(x_i)^\top(r_x - x_i) - y_i\nabla f_\theta(x_i)^\top(r_x - x_i) = (h'(f_\theta(x_i)) - y_i)\nabla f_\theta(x_i)^\top(r_x - x_i)$$

$$\psi_i''(0) = h'(f_\theta(x_i))(r_x - x_i)^\top\nabla^2 f_\theta(x_i)(r_x - x_i) + h''(f_\theta(x_i))[\nabla f_\theta(x_i)^\top(r_x - x_i)]^2$$
$$- y_i(r_x - x_i)^\top\nabla^2 f_\theta(x_i)(r_x - x_i).$$
$$= h''(f_\theta(x_i))\nabla f_\theta(x_i)^\top(r_x - x_i)(r_x - x_i)^\top\nabla f_\theta(x_i) + (h'(f_\theta(x_i)) - y_i)(r_x - x_i)^\top\nabla^2 f_\theta(x_i)(r_x - x_i)$$

By substituting these into equation 10 with $\varphi(\alpha) = \frac{1}{n}\sum_{i=1}^n \varphi_i(\alpha)$, we obtain the desired statement.

## A.2 Proofs related to adversarial robustness

### A.2.1 Proof of Lemma 3.2

Recall that $L_n^{adv}(\theta, S) = \frac{1}{n}\sum_{i=1}^n \max_{\|\delta_i\|_2 \leq \varepsilon\sqrt{d}} l(\theta, (x_i + \delta_i, y_i))$ and $g(u) = 1/(1 + e^{-u})$. Then the second-order Taylor expansion of $l(\theta, (x + \delta, t))$ is given by

$$l(\theta, (x + \delta, y)) = l(\theta, (x, y)) + (g(\theta^\top x) - y) \cdot \delta^\top\theta + \frac{1}{2}g(x^\top\theta)(1 - g(x^\top\theta)) \cdot (\delta^\top\theta)^2.$$

Consequently, for any given $\eta > 0$,

$$\max_{\|\delta\|_2 \leq \eta} l(\theta, (x + \delta, y)) = \max_{\|\delta\|_2 \leq \eta} l(\theta, (x, y)) + (g(\theta^\top x) - y) \cdot \delta^\top\theta + \frac{1}{2}g(x^\top\theta)(1 - g(x^\top\theta)) \cdot (\delta^\top\theta)^2$$

$$= l(\theta, (x, y)) + \eta|g(x^\top\theta) - y| \cdot \|\theta\|_2 + \frac{\eta^2}{2}(g(x^\top\theta)(1 - g(x^\top\theta))) \cdot \|\theta\|_2,$$

where the maximum is taken when $\delta = \text{sgn}(g(x^\top\theta) - y) \cdot \frac{\theta}{\|\theta\|} \cdot \eta$.

### A.2.2 Proof of Theorem 3.1

Since $f_\theta(x) = x^\top\theta$, we have $\nabla f_\theta(x_i) = \theta$ and $\nabla^2 f_\theta(x_i) = 0$. Since $h(z) = \log(1 + e^z)$, we have $h'(z) = \frac{e^z}{1+e^z} = g(z) \geq 0$ and $h''(z) = \frac{e^z}{(1+e^z)^2} = g(z)(1 - g(z)) \geq 0$. By substituting these into the equation of Lemma 3.1 with $\mathbb{E}_{r_x}[r_x] = 0$,

$$\tilde{L}_n^{\text{mix}}(\theta, S) = \tilde{L}_n^{\text{mix}}(\theta, S) + \mathcal{R}_1(\theta, S) + \mathcal{R}_2(\theta, S), \tag{11}$$

where

$$\mathcal{R}_1(\theta, S) = \frac{\mathbb{E}_\lambda[(1-\lambda)]}{n} \sum_{i=1}^{n} (y_i - g(x_i^\top \theta)) \theta^\top x_i$$

$$\mathcal{R}_2(\theta, S) = \frac{\mathbb{E}_\lambda[(1-\lambda)^2]}{2n} \sum_{i=1}^{n} |g(x_i^\top \theta)(1 - g(x_i^\top \theta))| \theta^\top \mathbb{E}_{r_x}[(r_x - x_i)(r_x - x_i)^\top]\theta$$

$$\geq \frac{\mathbb{E}_\lambda[(1-\lambda)]^2}{2n} \sum_{i=1}^{n} |g(x_i^\top \theta)(1 - g(x_i^\top \theta))| \theta^\top \mathbb{E}_{r_x}[(r_x - x_i)(r_x - x_i)^\top]\theta$$

where we used $\mathbb{E}[z^2] = E[z]^2 + \mathrm{Var}(z) \geq E[z]^2$ and $\theta^\top \mathbb{E}_{r_x}[(r_x - x_i)(r_x - x_i)^\top]\theta \geq 0$. Since $\mathbb{E}_{r_x}[(r_x - x_i)(r_x - x_i)^\top] = \mathbb{E}_{r_x}[r_x r_x^\top - r_x x_i^\top - x_i r_x^\top + x_i x_i^\top] = \mathbb{E}_{r_x}[r_x r_x^\top] + x_i x_i^\top$ where $\mathbb{E}_{r_x}[r_x r_x^\top]$ is positive semidefinite,

$$\mathcal{R}_2(\theta, S) \geq \frac{\mathbb{E}_\lambda[(1-\lambda)]^2}{2n} \sum_{i=1}^{n} |g(x_i^\top \theta)(1 - g(x_i^\top \theta))| \theta^\top (\mathbb{E}_{r_x}[r_x r_x^\top] + x_i x_i^\top)\theta.$$

$$\geq \frac{\mathbb{E}_\lambda[(1-\lambda)]^2}{2n} \sum_{i=1}^{n} |g(x_i^\top \theta)(1 - g(x_i^\top \theta))| (\theta^\top x_i)^2$$

$$= \frac{\mathbb{E}_\lambda[(1-\lambda)]^2}{2n} \sum_{i=1}^{n} |g(x_i^\top \theta)(1 - g(x_i^\top \theta))| \|\theta\|_2^2 \|x_i\|_2^2 (\cos(\theta, x_i))^2$$

$$\geq \frac{R^2 c_x^2 \mathbb{E}_\lambda[(1-\lambda)]^2 d}{2n} \sum_{i=1}^{n} |g(x_i^\top \theta)(1 - g(x_i^\top \theta))| \|\theta\|_2^2$$

Now we bound $E = \frac{\mathbb{E}_\lambda[(1-\lambda)]}{n} \sum_{i=1}^{n} (y_i - g(x_i^\top \theta))(\theta^\top x_i)$ by using $\theta \in \Theta$. Since $\theta \in \Theta$, we have $y_i f_\theta(x_i) + (y_i - 1) f_\theta(x_i) \geq 0$, which implies that $(\theta^\top x_i) \geq 0$ if $y_i = 1$ and $(\theta^\top x_i) \leq 0$ if $y_i = 0$. Thus, if $y_i = 1$,

$$(y_i - g(x_i^\top \theta))(\theta^\top x_i) = (1 - g(x_i^\top \theta))(\theta^\top x_i) \geq 0,$$

since $(\theta^\top x_i) \geq 0$ and $(1 - g(x_i^\top \theta)) \geq 0$ due to $g(x_i^\top \theta) \in (0, 1)$. If $y_i = 0$,

$$(y_i - g(x_i^\top \theta))(\theta^\top x_i) = -g(x_i^\top \theta)(\theta^\top x_i) \geq 0,$$

since $(\theta^\top x_i) \leq 0$ and $-g(x_i^\top \theta) < 0$. Therefore, for all $i = 1, \ldots, n$,

$$(y_i - g(x_i^\top \theta))(\theta^\top x_i) \geq 0,$$

which implies that, since $\mathbb{E}_\lambda[(1-\lambda)] \geq 0$,

$$\mathcal{R}_1(\theta, S) = \frac{\mathbb{E}_\lambda[(1-\lambda)]}{n} \sum_{i=1}^{n} |y_i - g(x_i^\top \theta)||\theta^\top x_i|$$

$$= \frac{\mathbb{E}_\lambda[(1-\lambda)]}{n} \sum_{i=1}^{n} |g(x_i^\top \theta) - y_i| \|\theta\|_2 \|x_i\|_2 |\cos(\theta, x_i)|$$

$$\geq \frac{R c_x \mathbb{E}_\lambda[(1-\lambda)]\sqrt{d}}{n} \sum_{i=1}^{n} |g(x_i^\top \theta) - y_i| \|\theta\|_2$$

By substituting these lower bounds of $\mathcal{R}_1(\theta, S)$ and $\mathcal{R}_2(\theta, S)$ into equation 11, we obtain the desired statement.

### A.2.3 PROOF OF THEOREM 3.2

Recall we assume that $\hat{\theta}_n$ will fall into the set $\Theta^*$ with probability at least $1 - \delta_n$, and $\delta_n \to 0$ as $n \to \infty$. In addition, define the set

$$\mathcal{X}_{\Theta^*}(\tau) = \{x \in \mathcal{X} : |\cos(x, \theta)| \geq \tau \text{ for all } \theta \in \Theta^*\},$$

there is $\tau \in (0, 1)$ such that $\mathcal{X}_{\Theta^*}(\tau) \neq \emptyset$, and

$$p_\tau := \mathbb{P}(x \in \mathcal{X}_{\Theta^*}(\tau)) \in (0, 1).$$

Let us first study

$$\frac{1}{n} \sum_{i=1}^n |g(x_i^\top \theta)(1 - g(x_i^\top \theta))|(\cos(\theta, x_i))^2$$

Since we assume $\Theta^*$ is bounded and $c_x \sqrt{d} \leqslant \|x_i\|_2 \leqslant b_x \sqrt{d}$ for all $i$, there exists $\kappa > 0$, such that $|g(x_i^\top \theta)(1 - g(x_i^\top \theta))| \geqslant \kappa$.

If we denote $\hat{p} = \{\text{number of } x_i's \text{ such that } x_i \in \mathcal{X}_{\Theta^*}(\tau)\}/n$. Then, it is easy to see

$$\frac{\frac{1}{n}\sum_{x_i \in \mathcal{X}_{\Theta^*}^c(\tau)} |g(x_i^\top \theta)(1 - g(x_i^\top \theta))|}{\frac{1}{n}\sum_{x_i \in \mathcal{X}_{\Theta^*}(\tau)} |g(x_i^\top \theta)(1 - g(x_i^\top \theta))|} \leqslant \frac{(1 - \hat{p})/4}{\hat{p}\kappa}$$

For $\eta^2$ satisfying

$$\eta^2 (1 + \frac{(1 - \hat{p})/4}{\hat{p}\kappa}) \leqslant \tau^2$$

we have

$$\frac{1}{n} \sum_{i=1}^n |g(x_i^\top \theta)(1 - g(x_i^\top \theta))|(\cos(\theta, x_i))^2 \geqslant \frac{1}{n} \sum_{x_i \in \mathcal{X}_{\Theta^*}(\tau)} |g(x_i^\top \theta)(1 - g(x_i^\top \theta))|\tau^2$$

$$\geqslant \frac{1}{n} \sum_{x_i \in \mathcal{X}_{\Theta^*}(\tau)} |g(x_i^\top \theta)(1 - g(x_i^\top \theta))|\eta^2 + \frac{1}{n} \sum_{x_i \in \mathcal{X}_{\Theta^*}^c(\tau)} |g(x_i^\top \theta)(1 - g(x_i^\top \theta))|\eta^2.$$

Lastly by Hoeffding's inequality, if we take $\varepsilon = p_\tau/2$

$$(1 + \frac{(1 - \hat{p})/4}{\hat{p}\kappa}) \leqslant (1 + \frac{(1 - p_\tau/2)/4}{(p_\tau/2)\kappa})$$

with probability at least $1 - 2\exp(-2n\varepsilon^2)$

$$\eta \leqslant \tau \sqrt{\frac{4\kappa p_\tau}{2 - p_\tau + 4\kappa p_\tau}}.$$

Similarly, if we study

$$\sum_{i=1}^n |g(x_i^\top \theta) - y_i||\cos(\theta, x)|$$

By boundedness of $\theta$, $x$ and $y \in \{0, 1\}$, we know there are constants $\kappa_1, \kappa_2 > 0$, such that

$$\kappa_1 \leqslant |g(f_\theta(x_i)) - y_i| \leqslant \kappa_2$$

Similarly, we know

$$\eta \leqslant \frac{p_\tau \kappa_1}{2\kappa_2 - p_\tau(\kappa_2 - \kappa_1)}\tau.$$

Combined together, we can obtain the result:

$$\eta \leqslant \min\{\frac{p_\tau \kappa_1}{2\kappa_2 - p_\tau(\kappa_2 - \kappa_1)}, \sqrt{\frac{4\kappa p_\tau}{2 - p_\tau + 4\kappa p_\tau}}\}\tau$$

### A.2.4 PROOF OF THEOREM 3.3

From the assumption, we have $f_\theta(x_i) = \nabla f_\theta(x_i)^\top x_i$ and $\nabla^2 f_\theta(x_i) = 0$. Since $h(z) = \log(1 + e^z)$, we have $h'(z) = \frac{e^z}{1+e^z} = g(z) \geq 0$ and $h''(z) = \frac{e^z}{(1+e^z)^2} = g(z)(1 - g(z)) \geq 0$. By substituting these into the equation of Lemma 3.1 with $\mathbb{E}_{r_x}[r_x] = 0$,

$$\tilde{L}_n^{\mathrm{mix}}(\theta, S) = \tilde{L}_n^{\mathrm{mix}}(\theta, S) + \mathcal{R}_1(\theta, S) + \mathcal{R}_2(\theta, S), \tag{12}$$

where

$$\mathcal{R}_1(\theta, S) = \frac{\mathbb{E}_\lambda[(1-\lambda)]}{n} \sum_{i=1}^n (y_i - g(f_\theta(x_i)))f_\theta(x_i)$$

$$\mathcal{R}_2(\theta, S) = \frac{\mathbb{E}_\lambda[(1-\lambda)^2]}{2n} \sum_{i=1}^n |g(f_\theta(x_i))(1 - g(f_\theta(x_i)))|\nabla f_\theta(x_i)^\top \mathbb{E}_{r_x}[(r_x - x_i)(r_x - x_i)^\top]\nabla f_\theta(x_i)$$

$$\geq \frac{\mathbb{E}_\lambda[(1-\lambda)]^2}{2n} \sum_{i=1}^n |g(f_\theta(x_i))(1 - g(f_\theta(x_i)))|\nabla f_\theta(x_i)^\top \mathbb{E}_{r_x}[(r_x - x_i)(r_x - x_i)^\top]\nabla f_\theta(x_i)$$

where we used $\mathbb{E}[z^2] = E[z]^2 + \mathrm{Var}(z) \geq E[z]^2$ and $\nabla f_\theta(x_i)^\top \mathbb{E}_{r_x}[(r_x - x_i)(r_x - x_i)^\top]\nabla f_\theta(x_i) \geq 0$. Since $\mathbb{E}_{r_x}[(r_x - x_i)(r_x - x_i)^\top] = \mathbb{E}_{r_x}[r_x r_x^\top - r_x x_i^\top - x_i r_x^\top + x_i x_i^\top] = \mathbb{E}_{r_x}[r_x r_x^\top] + x_i x_i^\top$ where $\mathbb{E}_{r_x}[r_x r_x^\top]$ is positive semidefinite,

$$\mathcal{R}_2(\theta, S) \geq \frac{\mathbb{E}_\lambda[(1-\lambda)]^2}{2n} \sum_{i=1}^n |g(f_\theta(x_i))(1 - g(f_\theta(x_i)))|\nabla f_\theta(x_i)^\top (\mathbb{E}_{r_x}[r_x r_x^\top] + x_i x_i^\top)\nabla f_\theta(x_i).$$

$$\geq \frac{\mathbb{E}_\lambda[(1-\lambda)]^2}{2n} \sum_{i=1}^n |g(f_\theta(x_i))(1 - g(f_\theta(x_i)))|(\nabla f_\theta(x_i)^\top x_i)^2$$

$$= \frac{\mathbb{E}_\lambda[(1-\lambda)]^2}{2n} \sum_{i=1}^n |g(f_\theta(x_i))(1 - g(f_\theta(x_i)))|\|\nabla f_\theta(x_i)\|_2^2 \|x_i\|_2^2 (\cos(\nabla f_\theta(x_i), x_i))^2$$

$$\geq \frac{R^2 c_x^2 \mathbb{E}_\lambda[(1-\lambda)]^2 d}{2n} \sum_{i=1}^n |g(f_\theta(x_i))(1 - g(f_\theta(x_i)))|\|\nabla f_\theta(x_i)\|_2^2$$

Now we bound $E = \frac{\mathbb{E}_\lambda[(1-\lambda)]}{n} \sum_{i=1}^n (y_i - g(f_\theta(x_i)))f_\theta(x_i)$ by using $\theta \in \Theta$. Since $\theta \in \Theta$, we have $y_i f_\theta(x_i) + (y_i - 1)f_\theta(x_i) \geq 0$, which implies that $f_\theta(x_i) \geq 0$ if $y_i = 1$ and $f_\theta(x_i) \leq 0$ if $y_i = 0$. Thus, if $y_i = 1$,

$$(y_i - g(f_\theta(x_i)))(f_\theta(x_i)) = (1 - g(f_\theta(x_i)))(f_\theta(x_i)) \geq 0,$$

since $(f_\theta(x_i)) \geq 0$ and $(1 - g(f_\theta(x_i))) \geq 0$ due to $g(f_\theta(x_i)) \in (0, 1)$. If $y_i = 0$,

$$(y_i - g(f_\theta(x_i)))(f_\theta(x_i)) = -g(f_\theta(x_i))(f_\theta(x_i)) \geq 0,$$

since $(f_\theta(x_i)) \leq 0$ and $-g(f_\theta(x_i)) < 0$. Therefore, for all $i = 1, \ldots, n$,

$$(y_i - g(f_\theta(x_i)))(f_\theta(x_i)) \geq 0,$$

which implies that, since $\mathbb{E}_\lambda[(1-\lambda)] \geq 0$,

$$\mathcal{R}_1(\theta, S) = \frac{\mathbb{E}_\lambda[(1-\lambda)]}{n} \sum_{i=1}^n |y_i - g(f_\theta(x_i))||f_\theta(x_i)|$$

$$= \frac{\mathbb{E}_\lambda[(1-\lambda)]}{n} \sum_{i=1}^n |g(f_\theta(x_i)) - y_i|\|\nabla f_\theta(x_i)\|_2 \|x_i\|_2 |\cos(\nabla f_\theta(x_i), x_i)|$$

$$\geq \frac{R c_x \mathbb{E}_\lambda[(1-\lambda)]\sqrt{d}}{n} \sum_{i=1}^n |g(f_\theta(x_i)) - y_i|\|\nabla f_\theta(x_i)\|_2$$

By substituting these lower bounds of $E$ and $F$ into equation 12, we obtain the desired statement.

## A.3 Proofs related to generalization

### A.3.1 Proof of Lemma 3.3 and Lemma 3.4

We first prove Lemma 3.3. The proof of Lemma 3.4 is similar.

By Eq. (9), we have $L_n^{\mathrm{mix}}(\theta, S) = L_n^{std}(\theta, \check{S})$, where $\check{S} = \{(\check{x}_i, y_i)\}_{i=1}^n$ with $\check{x}_i = \lambda x_i + (1 - \lambda)r_x$ and $\lambda \sim \tilde{\mathcal{D}}_\lambda = \frac{\alpha}{\alpha+\beta}Beta(\alpha+1, \beta) + \frac{\beta}{\alpha+\beta}Beta(\beta+1, \alpha)$. Since for Generalized Linear Model (GLM), the prediction is invariant to the scaling of the training data, so it suffices to consider $\tilde{S} = \{(\tilde{x}_i, y_i)\}_{i=1}^n$ with $\tilde{x}_i = \frac{1}{\lambda}(\lambda x_i + (1 - \lambda)r_x)$.

In the following, we analyze $L_n^{std}(\theta, \tilde{S})$. For GLM the loss function is

$$L_n^{std}(\theta, \tilde{S}) = \frac{1}{n}\sum_{i=1}^n l_{\tilde{x}_i, y_i}(\theta) = \frac{1}{n}\sum_{i=1}^n -(y_i\tilde{x}_i^\top\theta - A(\tilde{x}_i^\top\theta)),$$

where $A(\cdot)$ is the log-partition function in GLMs.

Denote the randomness (of $\lambda$ and $r_x$) by $\xi$, then the second order Taylor expansion yields

$$\mathbb{E}_\xi[A(\tilde{x}_i^\top\theta) - A(x_i^\top\theta)] \stackrel{2nd-order\ approx.}{=} \mathbb{E}_\xi[A'(x_i^\top\theta)(\tilde{x}_i - x_i)^\top\theta + A''(x_i^\top\theta)Var(\tilde{x}_i^\top\theta)]$$

Notice $\mathbb{E}_\xi[\tilde{x}_i - x_i] = 0$ and $Var_\xi(\tilde{x}_i) = \frac{1}{n}\sum_{i=1}^n x_i x_i^\top = \hat{\Sigma}_X$, then we have the RHS of the last equation equal to

$$A''(x_i^\top\theta)(\frac{\mathbb{E}(1-\lambda)^2}{\bar{\lambda}^2})\theta^\top\hat{\Sigma}_X\theta.$$

As a result, the second-order Taylor approximation of the Mixup loss $L_n^{std}(\theta, \tilde{S})$ is

$$\sum_{i=1}^n -(y_i x_i^\top\theta - A(x_i^\top\theta)) + \frac{1}{2n}[\sum_{i=1}^n A''(x_i^\top\theta)]\mathbb{E}(\frac{(1-\lambda)^2}{\lambda^2})\theta^\top\hat{\Sigma}_X\theta$$

$$= L_n^{std}(\theta, S) + \frac{1}{2n}[\sum_{i=1}^n A''(x_i^\top\theta)]\mathbb{E}(\frac{(1-\lambda)^2}{\lambda^2})\theta^\top\hat{\Sigma}_X\theta.$$

This completes the proof of Lemma 3.3. For Lemma 3.4, since the Mixup is performed on the final layer of the neural nets, the setting is the same as the least square with covariates $\sigma(w_j^\top x)$. Moreover, since we include both the linear coefficients vector $\theta_1$ and bias term $\theta_0$, the prediction is invariant to the shifting and scaling of $\sigma(w_j^\top x)$. Therefore, we can consider training $\theta_1$ and $\theta_0$ on the covariates $\{(\sigma(Wx_i) - \bar{\sigma}_W) + \frac{1-\lambda}{\lambda}(\sigma(Wr_x) - \bar{\sigma}_W)\}_{i=1}^n$, where $\bar{\sigma}_W = \frac{1}{n}\sum_{i=1}^n \sigma(Wx_i)$. Moreover, since we consider the least square loss, which is a special case of GLM loss with $A(u) = \frac{1}{2}u^2$, we have $A'' = 1$. Plugging these quantities into Lemma 3.3, we get the desired result of Lemma 3.4.

### A.3.2 Proof of Theorem 3.4 and Corollary 3.1

By definition, given $n$ $i.i.d.$ Rademacher rv. $\xi_1, ..., \xi_n$, the empirical Rademacher complexity is

$$Rad(\mathcal{W}_\gamma, S) = \mathbb{E}_\xi \sup_{a(\theta)\cdot\theta^\top\Sigma_X\theta\leq\gamma} \frac{1}{n}\sum_{i=1}^n \xi_i\theta^\top x_i$$

Let $\tilde{x}_i = \Sigma_X^{\dagger/2}x_i$, $a(\theta) = \mathbb{E}_x[A''(x^\top\theta)]$ and $v = \Sigma_X^{1/2}\theta$, then $\rho$-retentiveness condition implies $a(\theta)^2 \geq \rho \cdot \min\{1, \mathbb{E}_x(\theta^\top x)^2\} \geq \rho \cdot \min\{1, \theta^\top\Sigma_X\theta\}$ and therefore $a(\theta) \cdot \theta^\top\Sigma_X\theta \leq \gamma$ implies that $\|v\|^2 = \theta^\top\Sigma_X\theta \leq \max\{(\frac{\gamma}{\rho})^{1/2}, \frac{\gamma}{\rho}\}$.

As a result,

$$
\begin{aligned}
Rad(\mathcal{W}_\gamma, S) =& \mathbb{E}_\xi \sup_{a(\theta) \cdot \theta^\top \Sigma_X \theta \leq \gamma} \frac{1}{n} \sum_{i=1}^n \xi_i \theta^\top x_i \\
=& \mathbb{E}_\xi \sup_{a(\theta) \cdot \theta^\top \Sigma_X \theta \leq \gamma} \frac{1}{n} \sum_{i=1}^n \xi_i v^\top \tilde{x}_i \\
\leq& \mathbb{E}_\xi \sup_{\|v\|^2 \leq (\frac{\gamma}{\rho})^{1/2} \vee \frac{\gamma}{\rho}} \frac{1}{n} \sum_{i=1}^n \xi_i v^\top \tilde{x}_i \\
\leq& \frac{1}{n} \cdot (\frac{\gamma}{\rho})^{1/4} \vee (\frac{\gamma}{\rho})^{1/2} \cdot \mathbb{E}_\xi \| \sum_{i=1}^n \xi_i \tilde{x}_i \| \\
\leq& \frac{1}{n} \cdot (\frac{\gamma}{\rho})^{1/4} \vee (\frac{\gamma}{\rho})^{1/2} \cdot \sqrt{\mathbb{E}_\xi \| \sum_{i=1}^n \xi_i \tilde{x}_i \|^2} \\
\leq& \frac{1}{n} \cdot (\frac{\gamma}{\rho})^{1/4} \vee (\frac{\gamma}{\rho})^{1/2} \cdot \sqrt{\sum_{i=1}^n \tilde{x}_i^\top \tilde{x}_i} \,.
\end{aligned}
$$

Consequently,

$$
\begin{aligned}
Rad(\mathcal{W}_\gamma, S) = \mathbb{E}_S[Rad(\mathcal{W}_\gamma, S)] \leq& \frac{1}{n} \cdot (\frac{\gamma}{\rho})^{1/4} \vee (\frac{\gamma}{\rho})^{1/2} \cdot \sqrt{\sum_{i=1}^n \mathbb{E}_{x_i}[\tilde{x}_i^\top \tilde{x}_i]} \\
\leq& \frac{1}{\sqrt{n}} \cdot (\frac{\gamma}{\rho})^{1/4} \vee (\frac{\gamma}{\rho})^{1/2} \cdot rank(\Sigma_X).
\end{aligned}
$$

Based on this bound on Rademacher complexity, Corollary 3.1 can be proved by directly applying the following theorem.

**Lemma A.1** (Result from Bartlett & Mendelson (2002)). *For any $B$-uniformly bounded and $L$-Lipchitz function $\zeta$, for all $\phi \in \Phi$, with probability at least $1 - \delta$,*

$$
\mathbb{E}\zeta(\phi(x_i)) \leq \frac{1}{n} \sum_{i=1}^n \zeta(\phi(x_i)) + 2L Rad(\Phi, S) + B\sqrt{\frac{\log(1/\delta)}{2n}}.
$$

### A.3.3 PROOF OF THEOREM 3.5

To prove Theorem 3.5, by Lemma A.1, it suffices to show the following bound on Rademacher complexity.

**Theorem A.1.** *The empirical Rademacher complexity of $\mathcal{W}_\gamma^{NN}$ satisfies*

$$
Rad(\mathcal{W}_\gamma^{NN}, S) \leq 2\sqrt{\frac{\gamma \cdot (rank(\Sigma_X^\sigma) + \|\Sigma_X^{\sigma\dagger/2}\mu_\sigma\|^2)}{n}}.
$$

By definition, given $n$ *i.i.d.* Rademacher rv. $\xi_1, ..., \xi_n$, the empirical Rademacher complexity is

$$
Rad(\mathcal{W}_\gamma, S) = \mathbb{E}_\xi \sup_{\mathcal{W}_\gamma} \frac{1}{n} \sum_{i=1}^n \xi_i \theta_1^\top \sigma(W x_i).
$$

Let $\tilde{\theta}_1 = \Sigma_X^{\sigma 1/2}\theta_1$ and $\mu_\sigma = \mathbb{E}[\sigma(Wx)]$, then

$$
\begin{aligned}
\mathcal{R}_S(\mathcal{W}_\gamma^{NN}) =& \mathbb{E}_\xi \sup_{\mathcal{W}_\gamma^{NN}} \frac{1}{n} \sum_{i=1}^n \xi_i \tilde{\theta}_1^\top \Sigma_X^{\sigma\dagger/2}(\sigma(Wx_i) - \mu_\sigma) + \mathbb{E}_\xi \sup_{\mathcal{W}_\gamma^{NN}} \frac{1}{n} \sum_{i=1}^n \xi_i \tilde{\theta}_1^\top \Sigma_X^{\sigma\dagger/2}\mu_\sigma \\
\leq& \|\tilde{\theta}_1\|_2 \cdot \|\mathbb{E}_\xi[\frac{1}{n}\sum_{i=1}^n \xi_i \Sigma_X^{\sigma\dagger/2}\sigma(Wx_i)]\| + \|\tilde{\theta}_1\| \cdot \frac{1}{\sqrt{n}}\|\Sigma_X^{\sigma\dagger/2}\mu_\sigma\| \\
\leq& 2\sqrt{\frac{\gamma \cdot (rank(\Sigma_X^\sigma) + \|\Sigma_X^{\sigma\dagger/2}\mu_\sigma\|^2)}{n}},
\end{aligned}
$$

where the last inequality is obtained by using the same technique as in the proof of Lemma 3.4.

Combining all the pieces, we get

$$
Rad(\mathcal{W}_\gamma, S) \leq \sqrt{\frac{\gamma \cdot rank(\Sigma_X^\sigma)}{n}}.
$$

# B  DISCUSSION OF R IN THE NEURAL NETWORK CASE

**(B.1).** *On the value of $R = \min_i R_i$ via experiments for neural networks.* After training accuracy reaches 100%, the loss is further minimized when $\|f_\theta(x_i)\|_2$ increases. Since

$$
\|f_\theta(x_i)\|_2 = \|\nabla f_\theta(x_i)^\top x_i\|_2 = \|\nabla f_\theta(x_i)\|_2 \|x_i\|_2 R_i,
$$

this suggests that $R_i$ and $R$ tend to increase after training accuracy reaches 100%. We confirm this phenomenon in Figure 3. In the figure, $R$ is initially small but tend to increase after training accuracy reaches 100%, as expected. For example, for ANN, the values of $R$ were initially $2.27 \times 10^{-5}$ but increased to $6.11 \times 10^{-2}$ after training. Figure 3 (c) and (d) also show that $R_i$ for each $i$-th data point tends to increase during training and that the values of $R_i$ for many points are much larger than the pessimistic lower bound $R$: e.g., whereas $R = 6.11 \times 10^{-2}$, we have $R_i > 0.8$ for several data points in Figure 3 (d). For this experiment, we generated 100 data points as $x_i \sim \mathcal{N}(0, I)$ and $y_i = \mathbb{1}\{x_i^\top \theta^* > 0\}$ where $x_i \in \mathbb{R}^{10}$ and $\theta^* \sim \mathcal{N}(0, I)$. We used SGD to train linear models and ANNs with ReLU activations and 50 neurons per each of two hidden layers. We set the learning rate to be 0.1 and the momentum coefficient to be 0.9. We turned off weight decay so that $R$ is not maximized as a result of bounding $\|\nabla f_\theta(x_i)\|$, which is a trivial case from the above discussion.

**(B.2).** *A constant lower bound for neural networks.* Similarly, we can obtain a constant lower bound by adding some additional conditions.

**Assumption B.1.** *Let us denote $\hat{\Theta}_n \subseteq \Theta$ as the set of minimizers of $\tilde{L}_n^{mix}(\theta, S)$. We assume there exists a set $\Theta^*$, such that for all $n \geq N$, where $N$ is a positive integer, $\hat{\Theta}_n \subseteq \Theta^*$ with probability at least $1 - \delta_n$ and $\delta_n \to 0$ as $n \to 0$. Moreover, there exists $\tau, \tau' \in (0, 1)$ such that*

$$
\mathcal{X}_{\Theta^*}(\tau, \tau') = \{x \in \mathcal{X} : |\cos(x, \nabla f_\theta(x))| \geqslant \tau, \|\nabla f_\theta(x)\| \geqslant \tau', \text{ for all } \theta \in \Theta^*\},
$$

*has probability $p_{\tau,\tau'} \in (0, 1)$.*

**Theorem B.1.** *Define*

$$
\mathcal{F}_\Theta := \{f_\theta | f_\theta(x_i) = \nabla f_\theta(x_i)^\top x_i, \nabla^2 f_\theta(x_i) = 0 \text{ almost everywhere}, \theta \in \Theta\}.
$$

*Under Assumption B.1, for any $f_\theta(x) \in \mathcal{F}_\Theta$, if there exists constants $b_x, c_x > 0$ such that $c_x \sqrt{d} \leq \|x_i\|_2 \leq b_x \sqrt{d}$ for all $i \in \{1, \ldots, n\}$. Then, with probability at least $1 - \delta_n - 2\exp(-np_{\tau,\tau'}^2/2)$, there exist constants $\kappa > 0, \kappa_2 > \kappa_1 > 0$, if we further have $\theta \in \hat{\Theta}_n$, then*

$$
\tilde{L}_n^{mix}(\theta, S) \geq \frac{1}{n} \sum_{i=1}^n \tilde{l}_{adv}(\tilde{\varepsilon}_{mix}\sqrt{d}, (x_i, y_i))
$$

*where $\tilde{\varepsilon}_{mix} = \tilde{R}c_x \mathbb{E}_{\lambda \sim \tilde{\mathcal{D}}_\lambda}[1-\lambda]$ and $\tilde{R} = \min\{\sqrt{\frac{p_{\tau,\tau'}\kappa\tau'^2}{(2-p_{\tau,\tau'})/4\tau''^2 + p_{\tau,\tau'}\kappa\tau'^2}}, \frac{p_{\tau,\tau'}\kappa_1\tau'}{p_{\tau,\tau'}\kappa_1\tau' + (2-p_{\tau,\tau'})\kappa_2\tau''}\}\tau$.*

## B.1 PROOF OF THEOREM B.1

Notice if we assume for

$$\mathcal{X}_{\Theta^*}(\tau, \tau') = \{x \in \mathcal{X} : |\cos(x, \nabla f_\theta(x))| \geqslant \tau, \|\nabla f_\theta(x)\| \geqslant \tau', \text{ for all } \theta \in \Theta^*\},$$

there is $\tau, \tau' \in (0, 1)$ such that $\mathcal{X}_{\Theta^*}(\tau, \tau') \neq \emptyset$, and

$$p_{\tau,\tau'} := \mathbb{P}(x \in \mathcal{X}_{\Theta^*}(\tau, \tau')) \in (0, 1).$$

Let us first study

$$\frac{1}{n} \sum_{i=1}^n |g(f_\theta(x_i)) - y_i| \|\nabla f_\theta(x_i)\|_2 |\cos(\nabla f_\theta(x_i), x_i)|$$

By boundedness of $\theta$, $x$ and $y \in \{0, 1\}$, we know there is $\kappa_1, \kappa_2 > 0$, such that

$$\kappa_1 \leqslant |g(f_\theta(x_i)) - y_i| \leqslant \kappa_2$$

If we denote $\hat{p} = \{\text{number of } x_i's \text{ such that } x_i \in \mathcal{X}_{\Theta^*}(\tau, \tau')\}/n$. Then, it is easy to see

$$\frac{\frac{1}{n} \sum_{x_i \in \mathcal{X}_{\Theta^*}^c(\tau,\tau')} \|g(f_\theta(x_i)) - y_i\| \|\nabla f_\theta(x_i)\|_2}{\frac{1}{n} \sum_{x_i \in \mathcal{X}_{\Theta^*}(\tau,\tau')} |g(f_\theta(x_i)) - y_i| \|\nabla f_\theta(x_i)\|_2} \leqslant \frac{(1 - \hat{p})\kappa_2 \tau''}{\hat{p}\kappa_1 \tau'}$$

For $\eta^2$ satisfying

$$\eta(1 + \frac{(1 - \hat{p})\kappa_2 \tau''}{\hat{p}\kappa_1 \tau'}) \leqslant \tau$$

we have

$$\frac{1}{n} \sum_{i=1}^n |g(f_\theta(x_i)) - y_i| \|\nabla f_\theta(x_i)\|_2 \cos(\nabla f_\theta(x_i), x_i)| \geqslant \frac{1}{n} \sum_{i=1}^n |g(f_\theta(x_i)) - y_i| \|\nabla f_\theta(x_i)\|_2 \eta$$

Besides, if we consider

$$\sum_{i=1}^n |g(f_\theta(x_i))(1 - g(f_\theta(x_i)))| \|\nabla f_\theta(x_i)\|_2^2 (\cos(\nabla f_\theta(x_i), x_i))^2$$

Thus, we have

$$\eta^2(1 + \frac{(1 - \hat{p})/4\tau''^2}{\hat{p}\kappa\tau'^2}) \leqslant \tau^2$$

With probability at least $1 - 2\exp(-2n\varepsilon^2)$, for $\varepsilon = p_{\tau,\tau'}/2$, we have

$$\eta \leqslant \min\{\sqrt{\frac{p_{\tau,\tau'}\kappa\tau'^2}{(2 - p_{\tau,\tau'})/4\tau''^2 + p_{\tau,\tau'}\kappa\tau'^2}}, \frac{p_{\tau,\tau'}\kappa_1\tau'}{p_{\tau,\tau'}\kappa_1\tau' + (2 - p_{\tau,\tau'})\kappa_2\tau''}\}\tau$$

## B.2 PROOFS OF THE CLAIM $f_\theta(x) = \nabla f_\theta(x)^\top x$ AND $\nabla^2 f_\theta(x) = 0$ FOR NN WITH RELU/MAX-POOLING

Consider the neural networks with ReLU and max-pooling:

$$f_\theta(x) = W^{[L]}\sigma^{[L-1]}(z^{[L-1]}), \quad z^{[l]}(x, \theta) = W^{[l]}\sigma^{(l-1)}\left(z^{[l-1]}(x, \theta)\right), \, l = 1, 2, \ldots, L - 1,$$

with $\sigma^{(0)}\left(z^{[0]}(x, \theta)\right) = x$, where $\sigma$ represents nonlinear function due to ReLU and/or max-pooling, and $W^{[l]} \in \mathbb{R}^{N_l \times N_{l-1}}$ is a matrix of weight parameters connecting the $(l - 1)$-th layer to the $l$-th layer. For the nonlinear function $\sigma$ due to ReLU and/or max-pooling, we can define $\dot{\sigma}^{[l]}(x, \theta)$

such that $\dot{\sigma}^{[l]}(x,\theta)$ is a diagonal matrix with each element being 0 or 1, and $\sigma^{[l]}\left(z^{[l]}(x,\theta)\right) = \dot{\sigma}^{[l]}(x,\theta)z^{[l]}(x,\theta)$. Using this, we can rewrite the model as:

$$f_\theta(x) = W^{[L]}\dot{\sigma}^{[L-1]}(x,\theta)W^{[L-1]}\dot{\sigma}^{[L-2]}(x,\theta)\cdots W^{[2]}\dot{\sigma}^{[1]}(x,\theta)W^{[1]}x.$$

Since $\frac{\partial\dot{\sigma}^{[l]}(x,\theta)}{\partial x} = 0$ almost everywhere for all $l$, which will cancel all derivatives except for $\frac{d}{dx}W^{[1]}x$, we then have that

$$\frac{\partial f_\theta(x)}{\partial x} = W^{[L]}\dot{\sigma}^{[L-1]}(x,\theta)W^{[L-1]}\dot{\sigma}^{[L-2]}(x,\theta)\cdots W^{[2]}\dot{\sigma}^{[1]}(x,\theta)W^{[1]}. \tag{13}$$

Therefore,

$$\frac{\partial f_\theta(x)}{\partial x}x = W^{[L]}\dot{\sigma}^{[L-1]}(x,\theta)W^{[L-1]}\dot{\sigma}^{[L-2]}(x,\theta)\cdots W^{[2]}\dot{\sigma}^{[1]}(x,\theta)W^{[1]}x = f_\theta(x).$$

This proves that $f_\theta(x) = \nabla f_\theta(x)^\top x$ for deep neural networks with ReLU/Max-pooling.

Moreover, from equation 13, we have that

$$\nabla^2 f_\theta(x) = \nabla_x(W^{[L]}\dot{\sigma}^{[L-1]}(x,\theta)W^{[L-1]}\dot{\sigma}^{[L-2]}(x,\theta)\cdots W^{[2]}\dot{\sigma}^{[1]}(x,\theta)W^{[1]}) = 0,$$

since $\frac{\partial\dot{\sigma}^{[l]}(x,\theta)}{\partial x} = 0$ almost everywhere for all $l$. This proves that $\nabla^2 f_\theta(x) = 0$ for deep neural networks with ReLU/Max-pooling.

## C    MORE ABOUT EXPERIMENTS

### C.1    ADVERSARIAL ATTACK AND MIXUP

We demonstrate the comparison between Mixup and standard training against adversarial attacks created by FGSM. We train two WideResNet-16-8 (Zagoruyko & Komodakis, 2016) architectures on the Street View House Numbers SVHN (Netzer et al., 2011)) dataset; one model with regular empirical risk minimization and the other one with Mixup loss ($\alpha = 5, \beta = 0.5$). We create FGSM adversarial attacks (Goodfellow et al., 2014) for 1000 randomly selected test images. Fig. (1a) describes the results for the two models. It can be observed that the model trained with Mixup loss has better robustness.

### C.2    VALIDITY OF THE APPROXIMATION OF ADVERSARIAL LOSS

In this subsection, we present numerical experiments to support the approximation in Eq. (5) and (6). Under the same setup of our numerical experiments of Figure 2, we experimentally show that the quadratic approximation of the adversarial loss is valid. Specifically, we train a Logistic Regression model (as one example of a GLM model, which we study later) and a two layer neural network with ReLU activations. We use the two-moons dataset (Buitinck et al., 2013). Fig. 4, and compare the approximated adversarial loss and the original one along the iterations of computing the original adversarial loss against $\ell_2$ attacks. The attack size is chosen such that $\epsilon\sqrt{d} = 0.5$, and both models had the same random initialization scheme. This experiment shows that using second order Taylor expansion yields a good approximation of the original adversarial loss.

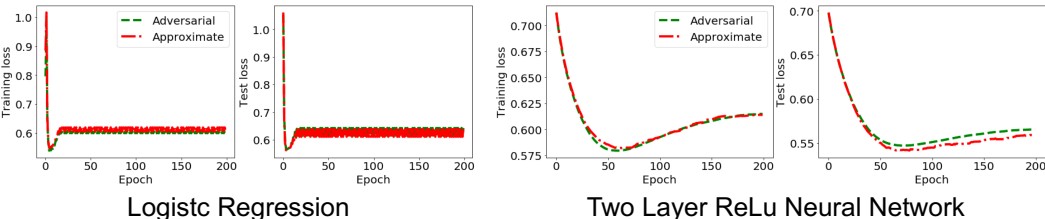

Figure 4: Comparison of the original adversarial loss with the approximate adversarial loss function.

### C.3 GENERALIZATION AND MIXUP

Figures 5–8 show the results of experiments for generalization with various datasets that motivated us to mathematically study Mixup. We followed the standard experimental setups without any modification as follows. We adopted the standard image datasets, CIFAR-10 (Krizhevsky & Hinton, 2009), CIFAR-100 (Krizhevsky & Hinton, 2009), Fashion-MNIST (Xiao et al., 2017), and Kuzushiji-MNIST (Clanuwat et al., 2019). For each dataset, we consider two cases: with and without standard additional data augmentation for each dataset. We used the standard pre-activation ResNet with 18 layers (He et al., 2016b). Stochastic gradient descent (SGD) was used to train the models with mini-batch size = 64, the momentum coefficient = 0.9, and the learning rate = 0.1. All experiments were implemented in PyTorch (Paszke et al., 2019).

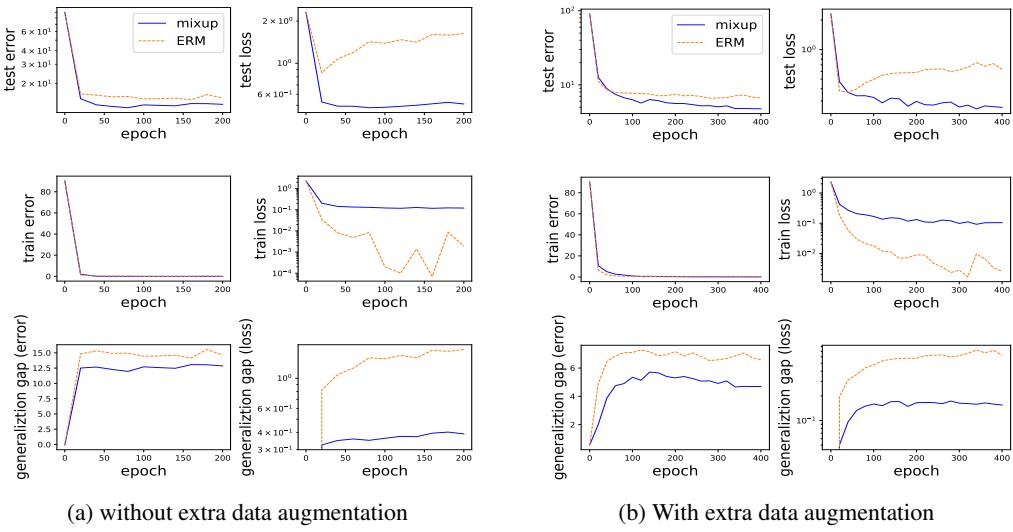

(a) without extra data augmentation      (b) With extra data augmentation

Figure 5: Generalization: CIFAR-10

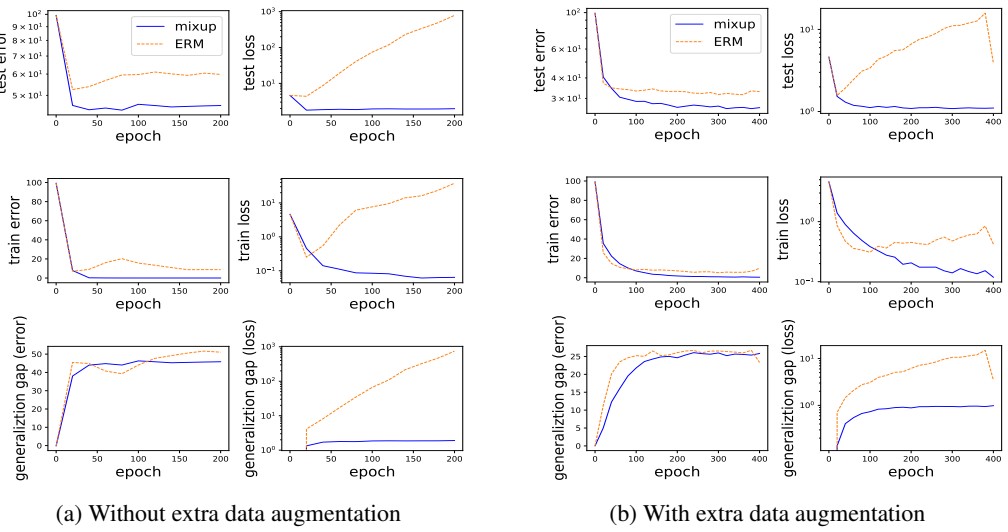

(a) Without extra data augmentation      (b) With extra data augmentation

Figure 6: Generalization: CIFAR-100

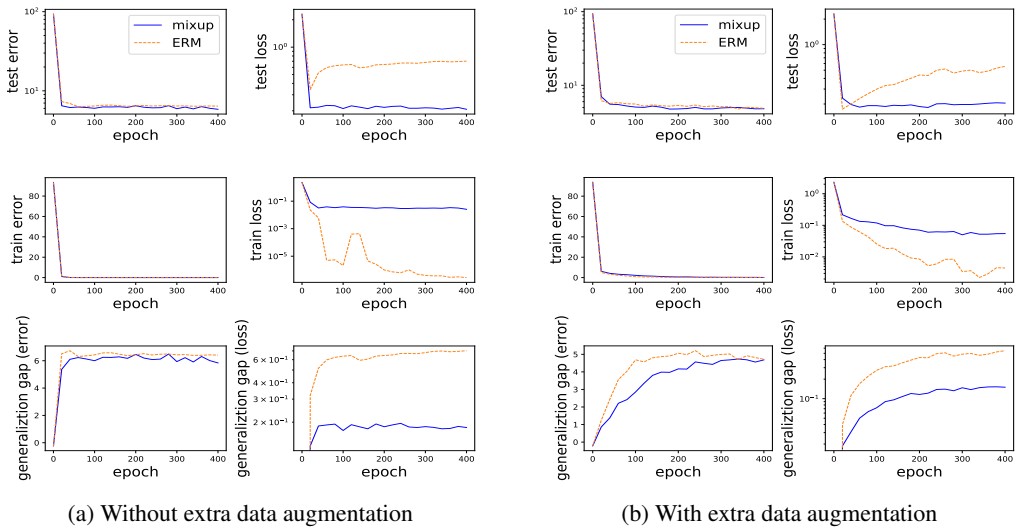

(a) Without extra data augmentation

(b) With extra data augmentation

Figure 7: Generalization: Fashion-MNIST

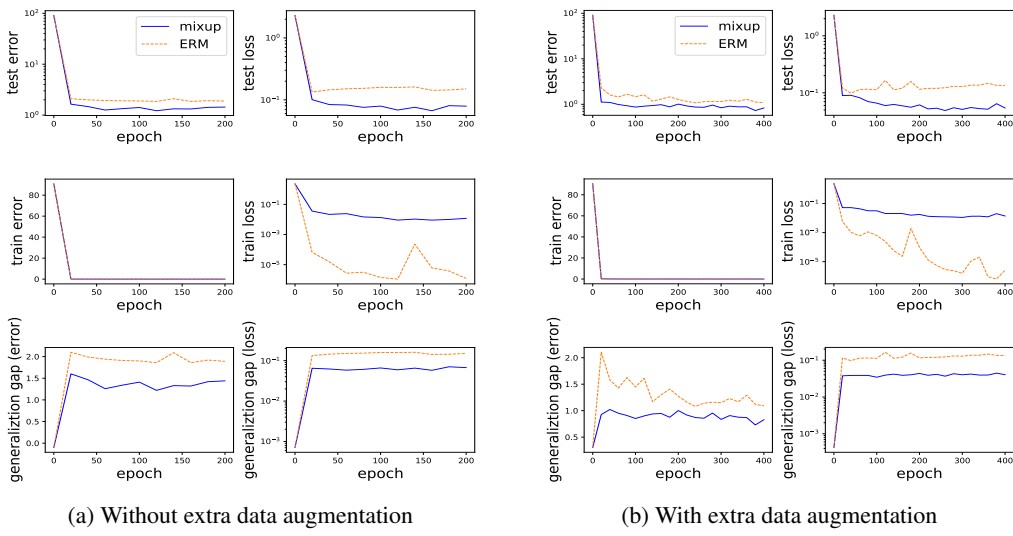

(a) Without extra data augmentation

(b) With extra data augmentation

Figure 8: Generalization: Kuzushiji-MNIST

