# OpenReview forum: "How Does Mixup Help With Robustness and Generalization?"
_ICLR.cc/2021/Conference — ICLR 2021 Spotlight_

### Official Review · AnonReviewer2 · 2020-10-26
**Careful analysis on theory, Good branch to help mixup community on adversarial robustness.**

**Rating:** 6
**Confidence:** 3

**Review:**

Summary

This paper has extensive analysis on mixup augmentation, which focus on the effect of adversarial robustness and generalization. In adversarial robustness, They try to make a connection between mixup loss and adversarial loss, On the other hand of generalization,  they argue that mixup is a kind of data-apdaptive regularization.

Comment
1. Good contribution about author's careful analysis on connecting between mixup loss and adversarial loss. It seems to be the first theoretical analysis on discussing their connection, since the past works just report the number to show how mixup and their variants to be robust to single-step adversarial attack.

2. Good to community about having a connection between Mixup and Rademacher complexity, I think it can make some impact to discuss the high-level connection between data augmentation and model generalization.

---

> ### Author Response · Authors · 2020-11-17
> **Response to Reviewer 2**
>
> Thank you for these kind comments! We are very excited about this work and the new theoretical insights it generates on the connection between Mixup and generalization and adversarial learning. For the high-level connection between data augmentation and model generalization, as have been shown in our paper, the Mixup training introduces an extra regularization term, which could help reduce the overfitting and lead to better generalization. We look forward to generalizing this technique to develop more connections between generalization performance and other data augmentation mechanisms.

---

### Official Review · AnonReviewer3 · 2020-10-28
**Good theoretical analysis**

**Rating:** 7
**Confidence:** 4

**Review:**

The paper theoretically studies the beneficial effect of mixup on robustness and generalization of machine models. The mixup loss is  rewritten to be the sum of the original empirical loss and a regularization term (plus a high order term). For robustness, the regularization term is proven to be upper bound of first and second order terms of the adversarial loss's Taylor expansion. Hence, the mixup loss can upper bound the approximate adversarial loss. For generalization, the regularization term is used to control the hypothesis to have small Rademacher complexity. The paper is clearly written and well organized.

pros:
1. Rigorous theoretical analysis are conducted on non-linear models, specifically the neural network model.
2. The theoretical results are clean and insightful.

cons:
1. When studying robustness, an approximated adversarial loss is considered. The approximated loss is the truncation of the Taylor expansion of the original loss. The quality of the approximation is not explored in the paper. It is better to provide numerical evidence that whether the bounds in Thm 3.1 and 3.3 still hold for original adversarial loss, and how tight the bounds are.
2. In the generalization part, only an indirect connection is built between mixup loss and the generalization gap. no result is provided concerning the generalization error of the solution found by minimizing the mixup loss.

---

> ### Author Response · Authors · 2020-11-18
> **Response to Reviewer 3**
>
> 1. On the numerical evidence on the approximation for original adversarial loss
>
> Response: Thank you for the kind and helpful suggestions. In light of your suggestions, we added two more numerical experiments (for logistic regression and two-layer NN with ReLU respectively) to show the approximation of adversarial loss by using second-order Taylor expansion. The results are added in Section C.2 ( in the revised appendix).
>
> 2. On the connection between the solution by directly minimizing the mixup loss and the results studied in Section 3.3
>
> Response: Thank you for your detailed comments. The indirect way by constructing connections between the regularized risk minimization and the constrained function class by the KKT conditions has been widely used in many previous machine learning works, for example, see [1,2,3]. To be more precise, a minimized regularized loss put a natural upper bound on the loss with the constraints in our paper by primal-dual formulation. In that way, we partially explain the phenomenon in our motivating experiments that Mixup helps generalization.
>
>
> [1]. Mohri, M., Rostamizadeh, A., and Talwalkar, A. Foundations of machine learning. MIT press, 2012.
>
> [2]. Sham M. Kakade, Karthik Sridharan, and Ambuj Tewari. On the complexity of linear prediction: Risk bounds, margin bounds, and regularization. In Proc. Neural Information Processing Systems, 2008.
>
> [3]. Raman Arora, Peter Bartlett, Poorya Mianjy, and Nathan Srebro. Dropout: Explicit forms and capacity control. arXiv preprint arXiv:2003.03397, 2020.

---

### Official Review · AnonReviewer1 · 2020-10-29
**A nice theoretical analysis on Mixup**

**Rating:** 7
**Confidence:** 4

**Review:**

This paper shows that Mixup training is approximately certain kind of regularized loss minimization. Based on this, it provides some theoretical analysis on the advantages of Mixup training for the generalization and adversarial robustness over one-step attacks.

This paper provides many insights on why Mixup works: e.g. connecting its 2nd order approximation with l2 adversarial loss; and shows that the Radmacher complexity of mixup adaptively characterize the intrinsic dimension of empirical data distribution. Though the techniques used in the paper were already developed by other works, the new results and insights on mixup in this work are worthy of being known by the community, particularly for that Mixup is such a popular data augmentation trick in deep learning.

Some questions:
1.	Could you provide any comments on Mixup and adversarial training, e.g. one-step and multi-step ones?
2.	What about the generality of the analysis on L_infinity attacks?

---

> ### Author Response · Authors · 2020-11-17
> **Response to Reviewer 1**
>
> 1. "Could you provide any comments on Mixup and adversarial training, e.g. one-step and multi-step ones?"
>
> Response: Thank you for this intriguing question. In our paper, we proved that the loss function induced by adversarial attacks is dominated by the loss function induced by Mixup up to the first two derivatives (we experimentally showed the accuracy of approximation).  As a result, this theorem suggests that Mixup training is approximately minimizing an upper bound of the first-order adversarial loss, and therefore partially justifies the numerical experiment in [1] that Mixup improves adversarial robustness against one-step attacks (FGSM). As a remark, we also would like to point out that unless further incorporating other techniques, Mixup training generally is not resistant to adversarial attacks with relatively large attack size generated by iterative methods such as projected gradient descent (PGD), which has also been experimentally explored in the same paper [1]. Meanwhile, the optimistic side is that Mixup is a popular data augmentation technique, and its vanilla version already demonstrates a certain degree of resistance to adversarial attacks while increasing the generalization performance at the same time. It would be interesting as a future research direction to see how incorporating other adversarial training methods would increase the overall adversarial robustness while not sacrificing too much accuracy. For example, in the paper [1], they showed combining Mixup and adversarial training will result in better robustness and standard accuracy. We have added this discussion in our updated version (on page 5).
>
> 2. "What about the generality of the analysis on L_infinity attacks?"
>
> Response: Thank you for this insightful question.  For the L_infinity attacks, we have
> max_{||delta||_ infty<epsilon} L < max_{||delta||_ 2<sqrt(d) epsilon} L,
> so the result shown in our paper still holds. If we want directly to solve this problem for L_infinity attacks,  when the attack is on L_infinity  norm, the problem is reduced to argmin_{||delta\|_infty<epsilon} (L+ \nabla L^T delta+ 1/2 delta \nabla^2 L delta). The minimization of the quadratic term is the Boolean quadratic programming (BQP) problem, which is NP-hard [2]. Thanks for mentioning this and we have added this discussion accordingly in the updated version (on page 5).
>
>
> [1] A. Lamb, V. Verma, J. Kannala, and Y. Bengio, “Interpolated adversarial training: Achieving robust neural networks without sacrificing too much accuracy,” in Proceedings of the 12th ACM Workshop on Artificial Intelligence and Security, ser. AISec’19, 2019, pp. 95–103.
>
> [2] Beasley, J. E. Heuristic algorithms for the unconstrained binary quadratic programming problem. London, England, 1998. 4.

---

### Official Review · AnonReviewer5 · 2020-11-05
**Good paper that gives theoretical guarantees about Mixup**

**Rating:** 8
**Confidence:** 3

**Review:**

**Paper summary**
The paper gives theoretical proof showing that the recently proposed data augmentation technique Mixup can indeed improve generalization and help in robustness. The theorems cover GLMs and certain classes of neural networks. The paper also contains numerical experiments supporting some aspects of the theory.

**Strengths**
1. Currently, there is only a limited theoretical understanding of why Mixup works. This paper shows that Mixup is essentially equal to regularizing the first and second derivatives (with respect to the input $x$). Intuitively, this means that changing the training samples slightly shouldn't change the output of the model much. Further, the paper proves that the mixup loss is an upper bound on the $2^{nd}$ order Taylor approximation of the adversarial loss, and hence reducing mixup loss reduces adversarial loss. Finally, the paper proves that mixup helps in reducing the Rademacher complexity and hence improves generalization.
2. The results seem fairly general and apply to many models such as GLMs and neural networks.
3. The paper supports its approximations and claims by numerical experiments.

**Concerns**
1. The regularizing term $\mathcal{R}_3$ looks like it is minimizing $z^T\nabla f_\theta(x_i) z$ (for some $z$). This promotes the Hessian (wrt $x$) to have negative eigenvalues in the direction of $z$. Ideally, we would want the Hessian (and also the gradient) to be 0 around the training samples so that perturbing the input doesn't change the output much. Thus, I don't see how the $\mathcal{R}_3$ term helps regularize the Hessian properly.
2. The paper claims that Assumption 3.1 holds when the minimizers are not too dispersed. Does it still hold for practical neural networks where the minimizers can possible be fairly far apart?


**Comments**
Although the paper seems well written, I have a few suggestions:
1. The notation $cos(\theta, x)$ which refers to $\frac{\langle \theta, x \rangle}{\|\theta\|\|x\|}$ should be explained in the preliminary section.
2. On page 6, the statement $f_\theta(x)=\nabla f_\theta(x)^Tx$ should be proven. It will save the reader some time if the proof is provided.
3. In Remark 3.1, I think Theorem 3.2 should actually be Theorem 3.4

**Score justification**
There isn't much prior work on the theoretical understanding of Mixup. This paper provides theoretical guarantees for Mixup on two fronts - robustness and generalization; for both GLMs and ReLUs.

---

> ### Author Response · Authors · 2020-11-17
> **Response to Reviewer 5**
>
> 1. On the regularizing term $R_3$
>
> Response: Thank you for this insightful question. The loss function and model family we considered follow the form  $l(\theta,(x,y))$=$h(f_{\theta}(x))$-$yf_{\theta}(x)$ for some function $h$, and $f_{\theta}$ is either a linear function (for GLMs) or neural network with ReLU/max pooling activation functions. Since $\theta^T x$, ReLU$(x)$, max-pooling$(x)$ all have zero second derivatives  (almost everywhere with respect to $x$), the term $\nabla_x^2f_\theta(x_i)$ would be zero for all $i$’s (almost everywhere with respect to $x_i$), which makes $R_3$ zero. Therefore, for these models, the issues you mentioned would not be a problem.  In addition, for more general cases, for activation functions similar to ReLU, such as softplus, $R_3$ will be very small. In broader scenarios beyond the model family we discussed, this problem is indeed worth further investigation.
>
> 2. On the Assumption 3.1
>
> Response: Thank you for this question. For the paragraph (after Assumption 3.1 on page 5) you mentioned, more precisely, it should be stated as that the minimizers are not located too dispersedly in the sense of angle (instead of Euclidean distance). In other words, if we normalize all the minimizers’ $\ell_2$ norm to one, this condition requires that the set of minimizers should not be located all over the sphere. Meanwhile, Assumption 3.1 only requires that the probability $p_\tau$ and the threshold $\tau$ to be non-zero, which we consider is a mild assumption. In particular, if the distribution of $x$ has positive masses in all solid angles, then when the set of minimizers is discrete, this assumption holds. For more complicated cases in which the set of minimizers consists of sub-manifolds, as long as there exists a solid angle in $\mathcal X$ that is disjoint with the set of minimizers, the assumption still holds. We have added this discussion accordingly in our revised paper (on page 6).
>
> Also, we would like to thank you for your comments regarding the presentation. We have corrected these typos in the updated version, and added the proof of $f_\theta(x) = \nabla_x f_\theta(x)^\top x$ and $\nabla_x^2f_\theta(x_i)=0$ (almost everywhere) in Section B.2 (in the revised appendix) according to your suggestions.

---

### Public Comment · ~Jy-yong_Sohn1 · 2020-11-17
**Minor comments**

I enjoyed reading this paper, and had some minor comments.

- In page 12, would you explain why $B\sim Bern(\frac{1}{2})$? Actually, I got $B\sim Bern(\frac{\alpha}{\alpha + \beta})$.
- Typo in page 13: A' & A'' -> h' & h''
- Typo in Lemma 3.1: Eq.(6) -> Eq.(1)

Overall, it was an interesting theory paper.

---

> ### Author Response · Authors · 2020-11-17
> **Thank you for your comments**
>
> Thank you for your comments, and we are glad that you enjoyed our paper! We have corrected all these typos in the updated version. Please see the updated version for details.

---

### Decision · Program_Chairs · 2021-01-07
**Final Decision**

**Decision:**

Accept (Spotlight)

**Comment:**

This paper provides theoretical justifications on why the data augmentation technique, Mixup (convex combinations of pairs of data examples) , can help in improving robustness and generalization of GLMs and ReLUs. The authors rewrote a Mixup loss function as the summation of a standard empirical loss and some regularization terms regularizing gradient, Hessian and some higher order terms. Using the quadratic approximation of the Mixup loss (ignoring the higher order terms), the authors proved that the quadratic approximation of the Mixup loss was equivalent to an upper bound of the second order Taylor expansion of an adversarial loss, providing justifications for why Mixup loss training could improve robustness against small attacks. Using the same quadratic approximation of the Mixup loss, the regularization term controlled the hypothesis class to have a smaller Rademacher complexity.

Overall, the paper provides insightful theoretical interpretations for a commonly used data augmentation technique in DL. The paper also supports its claims by numerical experiments. Although there is some minor concerns on using the quadratic approximation of the Mixup loss, as well as R3 term's regularization effect on a broader family of models, the paper provides unique and novel insights on Mixup.; all reviewers acknowledge the authors applying the existing proof techniques to analyze Mixup's effect on robustness and generalization.

Therefore, I recommend accepting this paper.